# Allostery can convert binding free energies into concerted domain motions in enzymes

Nicole Stéphanie Galenkamp [1,3], Sarah Zernia [1,3], Yulan B. Van Oppen[2], Marco van den Noort [1], Andreas Milias-Argeitis [2] & Giovanni Maglia [1] ✉

Enzymatic mechanisms are typically inferred from structural data. However, understanding enzymes require unravelling the intricate dynamic interplay between dynamics, conformational substates, and multiple protein structures. Here, we use single-molecule nanopore analysis to investigate the catalytic conformational changes of adenylate kinase (AK), an enzyme that catalyzes the interconversion of various adenosine phosphates (ATP, ADP, and AMP). Kinetic analysis validated by hidden Markov models unravels the details of domain motions during catalysis. Our findings reveal that allosteric interactions between ligands and cofactor enable converting binding energies into directional conformational changes of the two catalytic domains of AK. These coordinated motions emerged to control the exact sequence of ligand binding and the affinity for the three different substrates, thereby guiding the reactants along the reaction coordinates. Interestingly, we find that about 10% of enzymes show altered allosteric regulation and ligand affinities, indicating that a subset of enzymes folds in alternative catalytically active forms. Since molecules or proteins might be able to selectively stabilize one of the folds, this observation suggests an evolutionary path for allostery in enzymes. In AK, this complex catalytic framework has likely emerged to prevent futile ATP/ADP hydrolysis and to regulate the enzyme for different energy needs of the cell.

Enzymes evolved to stabilize the transition state of a reaction. However, enzymes designed on the bases of transition-state stabilization have shown to capture only a fraction of the catalytic efficiency of natural enzymes[1,2]. Additionally, transition state stabilization cannot explain why enzymes with identical active site structures show different catalytic efficiency at the same temperature[3], or why modifications far from the active site that do not affect the fold of the enzyme can have a large influence on catalysis[4–8]. At the same time, it is now accepted that proteins are intrinsically 'dynamic'. However, the role of dynamics in enzyme catalysis remains a topic of heated discussion[9]. It is possible, therefore, that enzyme structures and conformational dynamics evolved to have a more complex role than simply stabilize the transition state of a reaction.

In this work, we use single-molecule nanopore spectrometry[10–19] to monitor the enzyme adenylate kinase (AK). Compared to other techniques such as single-molecule FRET, nanopore spectrometry allows label-free sampling of the entire enzyme's dynamics during multiple turnovers of individual enzymes for minutes with microsecond resolution. AK catalyzes the reversible conversion of ATP and AMP to two molecules of ADP[20–23], which is vital to maintain cellular energy homeostasis[23–26]. The enzyme consists of a rigid core domain that holds the active site, the ATP-binding LID domain, and the AMP-binding NMP-domain (Fig. 1A)[27,28]. The LID and NMP domains undergo major conformation changes mainly induced by the binding of ATP/ADP/AMP. R123 and R156 in the LID domain, and R36 in the NMP domain are key residues for ligand binding in the closed configuration

[1]Chemical Biology I, Groningen Biomolecular Sciences & Biotechnology Institute, University of Groningen, 9747 AG Groningen, The Netherlands. [2]Molecular Systems Biology, Groningen Biomolecular Sciences & Biotechnology Institute, University of Groningen, 9747 AG Groningen, The Netherlands. [3]These authors contributed equally: Nicole Stéphanie Galenkamp, Sarah Zernia. ✉e-mail: giovanni.maglia@rug.nl

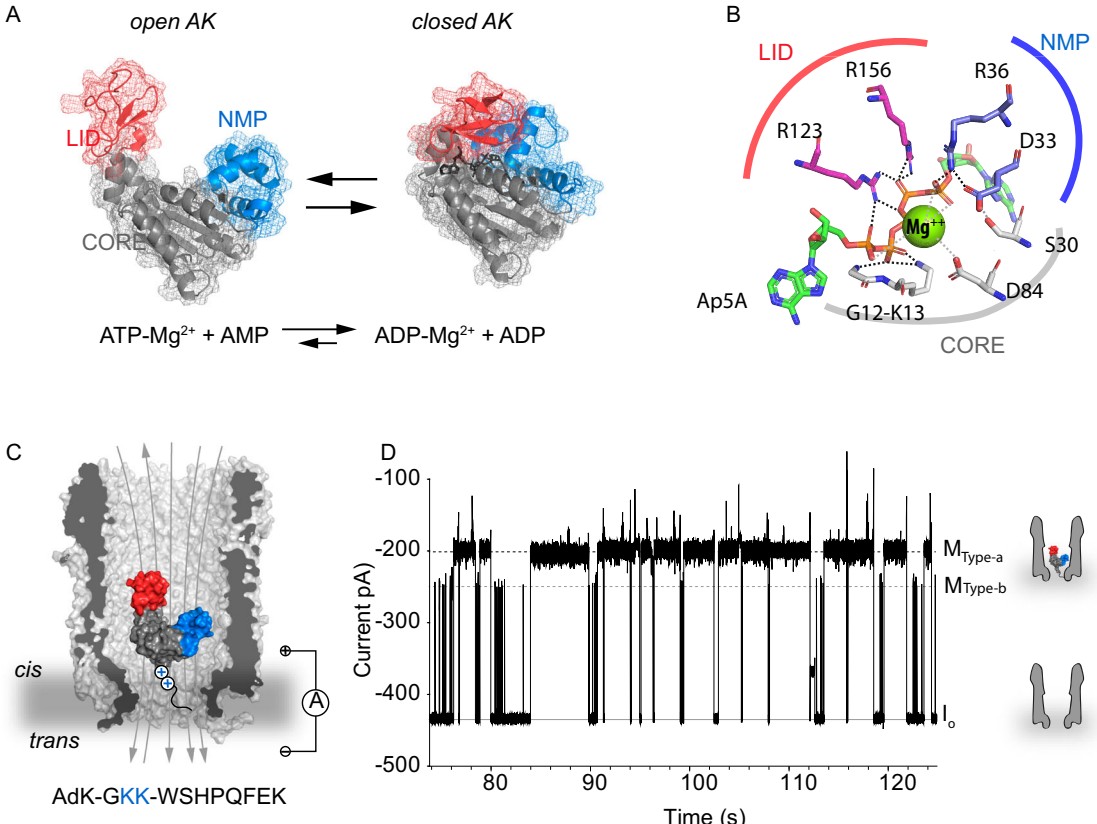

**Fig. 1 | Adenylate kinase inside the ClyA-AS nanopore. A** Crystal structure of *E. coli* adenylate kinase in open (PDB: 4ake) and closed conformation (Ap5A bound, PDB: 1ake). The enzyme is composed of a rigid core domain (CORE, grey) and two flexible domains that are closing upon binding of ATP, the LID domain (red, residues 121–161), and AMP, the NMP domain (blue, residues 3–60), respectively. **B** Residues within 6 Å from $Mg^{2+}$ in the closed AK bound to Ap5A (the coordinates for $Mg^{2+}$ were taken from PDB: 5G3Z). The dotted black lines indicate a binding distance of <3 Å, the dotted grey lines a distance >4 Å. The interactions between the three domains are all mediated by Ap5A. $Mg^{2+}$ is at binding distance only with the phosphate groups of Ap5A. **C** Schematic representation of AK, extended with two additional lysine residues and a Strep-tag at the C-terminus (AK_2 + )(Sequence depicted below), inside a ClyA-AS nanopore in a lipid bilayer (grey area). **D** Typical current trace after addition of 100 nM AK_2+ to a single ClyAS-AS nanopore in *cis* at −90 mV applied potential (*trans*). The grey line shows the open-pore current ($I_O$, −433.3 ± 5.7 pA), the black dashed line (M Type-a, −200.3 ± 2.4 pA) and the grey dashed line (M Type-b, −249.9 ± 3.7 pA) represent the blocked pore current. Error represents the standard deviation of the mean between independent experiments ($N$ = 3). The measurements were performed in 400 mM KCl, 15 mM Tris, 2 mM $MgCl_2$, pH 7.5 at room temperature (22 °C) applying −90 mV (*trans*) and sampling at 50 kHz with a 10 kHz Bessel-low filter, additionally digitally filtered with a 2 kHz Gaussian low-pass filter.

(Fig. 1B). Any mutation of these residues leads to substantially reduced enzyme activity[29–36]. A $Mg^{2+}$ cofactor ion is required for catalysis[29] and has been reported to influence domain motion[37]. The role of magnesium, however, remains enigmatic because despite coordinating the two nucleotides enabling phosphate transfer[38], it does not influence protein ligand affinity[39] nor interacts directly with either the NMP or LID domains in the closed state (Fig. 1B)[35].

Many studies have attempted to elucidate the role of domain motions in AK, and how dynamics and conformational changes are associated with molecular recognition, catalysis and allostery[40–56]. It has been described that the binding of ATP/ADP/AMP induces the closing of the domains. However, the nature of the motions of the two domains remains controversial, with some studies proposing that the LID domain closes first[38,51,53,54,56–67], others that the NMP domain closes first[58,64,65,67,68] and others that both domain close simultaneously[49,55,69].

Our results showed that the LID and NMP domain close in a precise sequence, which allows regulating the enzyme's affinity and binding hierarchy for ATP, ADP, and AMP. A detailed kinetic analysis and modeling revealed a sophisticated mechanism in which the enzymatic function is regulated by multiple allosteric interactions that modulate the entire collection of enzyme dynamics.

## Results

### Protein trapping inside ClyA nanopres

Adenylate kinase from *E. coli* (Fig. 1A) added to the *cis* side of a ClyA nanopore (Fig. 1C) was trapped by the action of the electroosmotic flow[70] generated by applying negative transmembrane potentials (*trans*). Wild-type adenylate kinase from *E. coli* (Fig. 1A) bears a net negative charge at pH 7.5 (PI = 5.5), which prevented its efficient trapping inside the nanopore. Therefore, the enzyme was extended with two lysine residues just before the purification tag (Strep-tag) at the C-terminus (AK_2 + , hereafter simply referred to as AK). These modifications only slightly affected the activity of the enzyme (Supplementary Fig. 1, Supplementary Table 1), while also enabling longer protein trapping times, most likely the result of the protein aligning with the electric field inside the nanopore. The protein entered the nanopore into more than one orientation (named Type-a, Type-b and Type-c, Fig. 1D, Supplementary Fig. 2, supporting information for additional discussion). Here, we focus on Type-a blockades, which had a residual current [$I_{RES\%}$, defined as the percent ratio between the protein blocked current ($I_B$) and the open pore current ($I_O$)] of 46.0 ± 0.1 %. At −90 mV applied potential, Type-a produced low-noise and second-long signals (Fig. 1D, Fig. 2A, Supplementary Figs. 2, 3, 4).

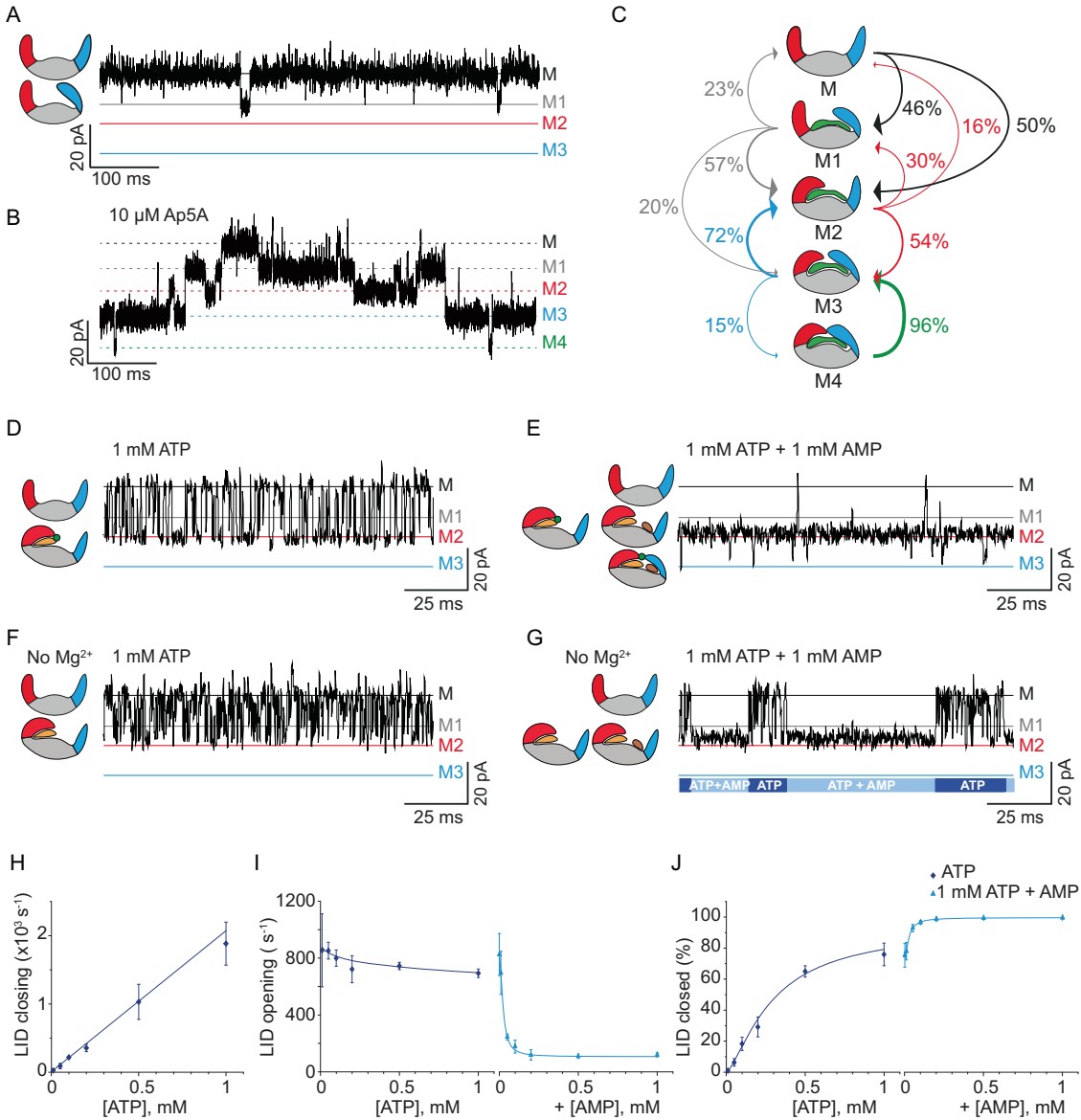

**Fig. 2 | AK_2+ current blockades in the ClyA-AS nanopore upon binding of Ap5A, ATP, and AMP. A** Expansion of a typical blockade induced by the apo-AK in ClyA-AS. **B** A typical Ap5A ligand-induced events showing four additional levels (M1-M4). **C** Scheme indicating the transitions observed with Ap5A and their molecular interpretation. Typical blockades in the presence of 1 mM ATP (**D**), 1 mM ATP and 1 mM AMP (**E**), 1 mM ATP and no Mg²⁺ in solution (**F**), and 1 mM ATP and 1 mM AMP with no Mg²⁺ (**G**). **H** Frequency of LID closing - measured as the inverse of dwell times – at increasing ATP concentrations. The line indicates a linear fit. **I** LID opening frequency at increasing ATP concentrations. The curves are fits of the data to a Hill function. **J** Percentage of the closed LID domain at increasing ligand concentrations. The curves correspond to fits of a Hill function. The measurements were performed in 400 mM KCl, 15 mM Tris, 2 mM MgCl₂ (except for panels F and G), pH 7.5 at room temperature (22 °C) applying −90 mV (*trans*) and sampling at 50 kHz with a 10 kHz Bessel-low filter, additionally digitally filtered with a 2 kHz Gaussian low-pass filter. Error bars represent the standard deviation of the mean between independent experiments (*N* = 3). Ligands were added to the *trans* chamber and the enzyme to the *cis* chamber.

## LID and NMP domain motions induced by ligands

The conformational changes of AK were sampled using $P^1,P^5$-Di-(adenosine-5')-pentaphosphate (Ap5A), an inhibitor that is known to induce the closing of AK LID and NMP domains into more than one possible conformation[28,34,47,66]. The apo-enzyme showed one main current level that occasionally switched to a deeper state (M1, $I_{RES}^{M1} = 49.2 \pm 0.1$ %, $k_{on}^{M1} = 4.66 \pm 0.72$ s⁻¹ and $k_{off}^{M1} = 55.2 \pm 15.8$ s⁻¹, Fig. 2A, Supplementary Fig. 3). Previous work indicated that the enzyme might spontaneously close[34,42,48,49,54,63,71–74], suggesting that M1 might indicate the LID or NMP domain closing without any ligand present. The addition of Ap5A to either the *cis* or *trans* side of the nanopore induced four interconverting current levels: M1, M2, M3 and M4 ($I_{RES\_M1} = 48.7 \pm 0.2$ %,

$I_{RES\_M2} = 51.0 \pm 0.1$ %, $I_{RES\_M3} = 53.8 \pm 0.1$ %, $I_{RES\_M4} = 56.6 \pm 0.2\%$, respectively, Fig. 2B, C, Supplementary Figs. 4, 5, Supplementary Table 2), most likely, representing all possible domain configurations that AK can adopt within the nanopore.

ATP added to the *trans* chambers induced M2 current levels ($I_{RES\_M2} = 52.0 \pm 0.1\%$, Fig. 2D, Supplementary Fig. 8, Supplementary Table 2). The frequency of M2 blockades increased linearly with the ATP in solution ($k_{on}^{M2} = 2.2 \pm 0.08$ μM⁻¹ s⁻¹), and the dwell time remained constant (average $k_{off}^{M2} = 800 \pm 57$ s⁻¹ at 100 μM ATP, Fig. 2H, I). Consequentially, M2 blockades most likely reflect the closing of the LID domain around ATP. Consistently with previous reports[42,47,75], at saturating concentrations of ATP (1 mM), only $75.9 \pm 7.3$ % of AK was

closed (Fig. 2J, Supplementary Table 3), indicating that the binding of ATP to AK does not always trigger the closing of the LID domain. In the absence of the cofactor $Mg^{2+}$, similar M2 blockades were observed, although the affinity of ATP for AK was reduced due to a higher $k_{off}^{M2}$ (Fig. 2F, Supplementary Fig. 9, Supplementary Fig. 10), indicating that the cofactor is involved in stabilizing the domain in the closed configuration. By contrast, AMP in solution induced only very few additional current blockades (Supplementary Fig. 6, Supplementary Fig. 7), suggesting that domain closure requires higher adenosine phosphates concentrations, in line with previous reports[34]. Therefore, M2 most likely represents the LID domain motions as measured by single-molecule FRET[50] or AFM[47]. Notably, the μs domain motions, as observed by FRET, [51] are likely too fast to be observed here.

The subsequent addition of 1 mM AMP (trans) in the presence of 1 mM ATP (trans) decreased the M2 → M rate by ~7 folds (119.6 ± 13.0 s$^{-1}$, Fig. 2E, I, Supplementary Fig. 11). Under these conditions, the probability of observing a closed LID domain increased to nearly 100% (99.5 ± 0.1 %, Fig. 2E, J, Supplementary Fig. 12, Supplementary Table 5). Previous work also showed that AMP increased the closed probability in the presence of ATP, although to a smaller degree[50]. These data indicate that AMP enhances the likelihood of the LID domain to close, the hallmark of allostery. Here, we will use the term endo-allostery or endostery to indicate when allostery is mediated by the reactants themselves rather than by an effector molecule, and it differs from cooperativity as the two reactants bind to the same active site. Long M2 (40.3 ± 18.0 s$^{-1}$) events were also observed in the absence of $Mg^{2+}$ (Fig. 2G, Supplementary Fig. 13, Supplementary Table 5), indicating that the AMP-induced effect is not mediated by the cofactor $Mg^{2+}$.

When AMP was added in the presence of ATP, M3 events were observed ($I_{RES\_M3}$ = 54.5 ± 0.2 %, Fig. 2E, Supplementary Figs. 12, 14, Supplementary Table 2, Supplementary Table 5). However, in the absence of $Mg^{2+}$ (Fig. 2G, Supplementary Fig. 13), or when sampling the inactive mutant R156A-AK (Supplementary Figs. 15–17), M3 events were drastically reduced or not observed. In R156A-AK the interaction between the ATP and the LID domain, and between the LID and NMP

domain is impaired[31,76]. Therefore, M3 most likely reflects the closing of the NMP domain over the AMP:ATP:AK complex, whereby the LID domain is already closed. Notably, M3 events occurred from the M2 level, indicating that the NMP domain can only close after the LID domain is already closed.

The M1 and M4 current levels observed in the Ap5A blockades may also be explained as domain movements (Fig. 2B, C, Supplementary Fig. 5). M4 is only observed when Ap5A is added to the solution and is likely to represent the Ap5A-induced fully closed enzyme as observed in the crystal structures and in previous AFM experiments[28,45,47]. M1 cannot be easily assigned because it is not directly correlated to the binding of any one ligand. Likely, M1 reflects the closing of only one domain, the NMP domain, while the other domain, the LID domain, is still open. Interestingly, M1 to M2 transitions were often observed when both binding sites are occupied i.e.; when ATP and AMP (Supplementary Fig. 11) or Ap5A (Supplementary Fig. 5) are sampled, indicating that when both active sites are occupied the two domains cooperatively open and close. As previously suggested for the observed fast LID domain fluctuations[48], these concerted motions might be instrumental to align the reactants in the proper conformation. The assignment of the four M states is compatible with a physical model for protein-nanopore interactions, which proposes that when the protein's structure becomes more compacted, it penetrates deeper inside the nanopore, resulting in a higher current block[14]. A full kinetic scheme for the binding of ATP and AMP is shown in Fig. 3.

### Allosteric effects and concerted domain motions are linked to the catalytic step

The ligand induced domain transitions of AK followed a well-defined hierarchy. In the absence of ATP, AMP was not observed to induce the closing of the NMP domain. By contrast, ATP induces the closing of the LID domain, and the subsequent arrival of AMP induces the closing of the NMP domain. These observations can be explained by the existence of an endosteric effect where the initial binding of AMP inhibits

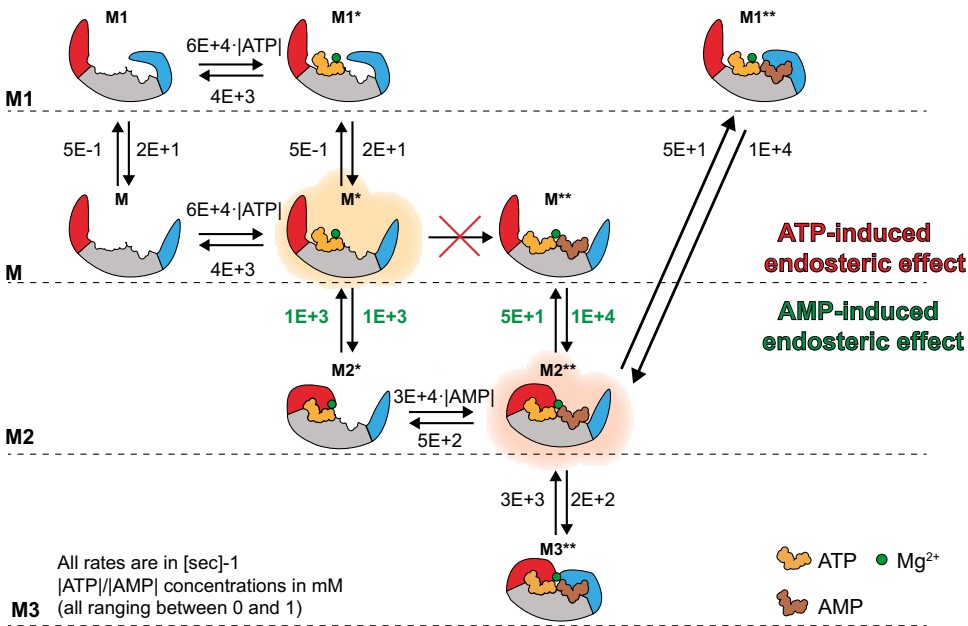

**Fig. 3 | Kinetic model for the endosteric hierarchical closing of the LID and NMP domain.** The model includes the four observed states for the LID and NMP domain, and the hidden ligand-bound states. The ATP-induced endosteric effect was implemented by removing the M* → M** transition. The AMP-induced endosteric effect was implemented by allowing the M* ↔ M2* and M** ↔ M2** states to have

different rates. The rates were retrieved by fitting the kinetic data to a Hidden Markov Model (HMM) based on the four observable states and using different ligand concentration. Confidence intervals for the estimated rates are given in Supplementary Table 7. The asterisks indicate whether one (*) or two (**) ligands are bound.

the binding of ATP by forming a less active conformation, as previously suggested[48], or the ATP-induced closing of the LID domain induces the formation a higher-affinity binding site for AMP. In the M2-ternary-AMP:ATP:AK complex, the LID domain is more tightly closed (Fig. 2I, right) than in the M2-binary-ATP:AK complex (LID opening of $118.9 \pm 36.3$ s$^{-1}$ and $693.4 \pm 29.6$ s$^{-1}$, respectively, Fig. 2I, left). Since under these conditions AMP does not close the LID domain, the increased LID closing is the result of a second endosteric rearrangement.

This catalytic mechanism was validated by constructing a Hidden Markov Model (HMM) based on the four experimentally observed conformational states (Figs. 2 and 3), where M is the open state, M1 the closed NMP domain, M2 the closed LID domain and M3 the LID and NMP both closed. We assumed that the binding of ATP and AMP is experimentally not observable by current recordings, but reversible binding of the ligands was included in the model, giving rise to nine hidden states (Fig. 3). We then introduced the minimal number of state transitions that allowed the model to reproduce the conformational transitions observed in our experiments, allowing the possibility that certain transitions take place with or without ligands.

To verify the presence of the endosteric effects described above, we compared three different model variants, each encoding a different set of assumptions that eliminated each of the endosteric effects. In model variant V1, the LID domain dynamics were independent of the presence of AMP, i.e., transitions M* ↔ M2* and M** ↔ M2** proceeded at the same rates. In model variant V2, binding of AMP was independent of the LID domain status, i.e., transitions of M* ↔ M** and M2* ↔ M2** had the same rates. Finally, model variant V3 encoded the presence of the two endosteric effects: the ATP-induced effect was implemented by eliminating the transition M* ↔ M**, and the AMP-induced effect by allowing the transitions M* ↔ M2* and M** ↔ M2** to proceed at different rates. The three model variants were then fitted to experimental data obtained at different ATP and AMP concentrations using maximum likelihood estimation (See Methods). Comparing the maximal likelihood values attained by the three models, we observed that the likelihood of V3 is overwhelmingly greater than the likelihood of V1 and V2 (Supplementary Fig. 18, Supplementary Table 9). Notably, the rates of LID domain closing/opening at different ATP/AMP concentrations (transition rates between groups of states) predicted by model V3, were also very close to the experimentally determined ones (Supplementary Fig. 18, Supplementary Table 10), further suggesting that this model captures essential aspects of the underlying system kinetics.

## ADP also triggers the same endosteric modulation and domain motions as ATP and AMP

When ADP was added to the *trans* chambers, M2 current levels were also observed ($I_{RES\_M2} = 51.8 \pm 0.2$ %) (Supplementary Table 2), indicating that ADP also induces the closing of the LID domain. Increasing the concentration of ligands increased the frequency of M2 events ($k_{on}^{M2} = 6.6 \pm 3.5 \, \mu M^{-1}$ s$^{-1}$, Fig. 4A, C, Supplementary Fig. 19, Supplementary Table 6). The $k_{off}^{M2}$ was concentration dependent, decreasing from $806 \pm 136$ s$^{-1}$ at 10 $\mu$M to $65.9 \pm 22.0$ s$^{-1}$ at 1 mM ADP (Fig. 4D, Supplementary Fig. 20), suggesting that, as observed before for the binding of AMP and ATP, the binding of a second ADP molecule to the AMP-binding site increased the probability for LID domain closure. Fitting a Hill function to the fraction of closed enzyme as a function of ligand concentration (Fig. 4E), revealed a K$_D$ of $45.4 \pm 1.6$, which is in the same range as those measured in bulk ensemble experiments (Supplementary Table 7); and the Hill coefficient was 2.3, which is compatible with an endosteric effect. The presence of the endosteric effect was also supported by our mathematical model, as shown in Supplementary Figs. 21 and 22 (Supplementary Table 12). LID closure could also be triggered in the absence of Mg$^{2+}$ (Fig. 4A, Supplementary Figs. 23, 24) and by AMP (Fig. 4E, Supplementary Figs. 25, 26), further

suggesting that it is not mediated by the interaction of Mg$^{2+}$ with the phosphate groups of the reactants. M3 events were also observed ($I_{RES\_M3} = 54.3 \pm 0.2$ and $53.5 \pm 0.8$ %, $k_{off} = 3.4 \pm 0.7 \times 10^3$, Fig. 4A, Supplementary Fig. 20, Supplementary Table 2, Supplementary Table 6), but only when Mg$^{2+}$ was present in solution (Fig. 4A, Supplementary Fig. 23). Together, these results indicate that, as observed for ATP and AMP, the binding of ADP to AK triggers a similar set of allosteric events that induce the concerted closure of the LID and NMP domains, with Mg$^{2+}$ playing a pivotal role.

## A subset of AK shows reduced endosteric control of domain closure

Unexpectedly, we observed a second kind of protein blockade, a subset of type-a blockades that we named type II, which comprised $12 \pm 6$ % of total blockades under 1 mM ADP. Interestingly, two native forms of AK with distinct kinetics properties have also been observed for Rabbit Muscle AK[77]. The observed blockades showed a similar M2 residual current level and on rate (Supplementary Table 2, Supplementary Table 6) compared to type I blockades, but a ~ 4 fold increased off rate (Fig. 4B, D, Supplementary Figs. 27, 28, Supplementary Table 6). The off rate decreased at higher ADP concentrations and during AMP addition (Fig. 4B, E, Supplementary Figs. 25, 27–29), but only slightly compared to type I blockades, suggesting that the binding of a second ADP molecule induced a weaker allosteric change within the enzyme. The K$_D$ of type II blockades increased to $123.2 \pm 18.1 \, \mu$M, and the Hill coefficient was reduced to 1.7, while the $k_{off}$ of M3 blockades remained unchanged ($3.2 \pm 0.3 \times 10^3$ s$^{-1}$, Fig. 4B, D, Supplementary Fig. 28, Supplementary Table 6).

## Discussion

In this work, we studied the conformational changes of AK using nanopore technology. This approach allows sampling the global dynamics of the enzyme with high-microsecond resolution for extensive time. We observed three independent LID-NMP domain motions in AK. By comparison, FRET could only measure the motions of the LID domain[50], while AFM[47] and NMR[45] measured the combination of both domains[31].

The ability to measure both LID and NMP domain motions independently revealed a precise hierarchy of domain closure during AK catalysis, controlled by endosteric interactions. The initial binding of ATP or ADP induces the closing of the LID domain at a rate of 1000 s$^{-1}$, which matches the domain closing rate that was measured by NMR for the combined NMP + LID domain motions ($1374 \pm 110$ s$^{-1}$)[45]. The LID domain may quickly reopen with a similar rate (1000 s$^{-1}$, Fig. 3). However, if another adenosine phosphate subsequently binds, it induces an endosteric effect that retards the opening of the LID domain and allows the additional closing of the NMP domain (Fig. 2J). The closing of the NMP domain has the slowest observed rate (200 s$^{-1}$), which matches the catalytic turnover rate of the enzyme ($263 \pm 30$ s$^{-1}$)[45]. Hence, the closing of the NMP domain is likely the rate-limiting step of the catalytic reaction. An early NMR study measured an opening rate for the combined NMP + LID domains of $286 \pm 85$ s$^{-1}$, and suggested that the rate-determining step is the enzyme opening after the catalytic reaction[29,45]. Possibly, this discrepancy arises from the simplistic two-state conformational transition model used in the NMR experiments for NMP + LID domain dynamics. A more complex model for AK domain dynamics has also been suggested by a more recent $^1$H–$^{15}$N HSQC NMR study, arguing that the chemical step is rate-determining[36]. However, also here, a two-state conformational transition model was used.

Notably, recent experimental and simulation findings[31,36,50,56,78,79] have reported LID domain motions on the order of microseconds for both the apo- and liganded enzyme (a rate of 22,000 s$^{-1}$)[50]. These large-scale LID-domain conformational dynamics, which might be required to position the substrates for their optimal orientation during

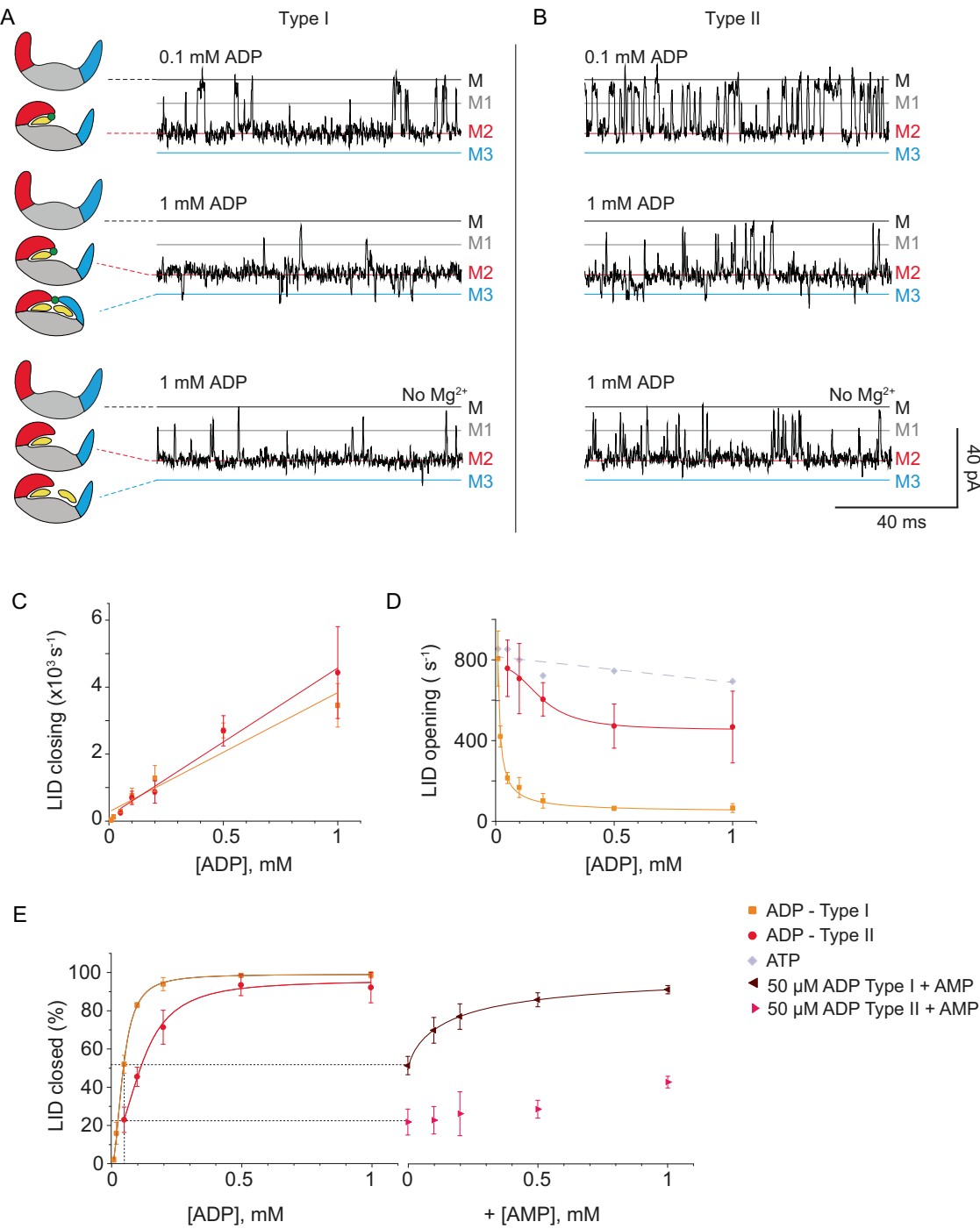

**Fig. 4 | AK current blockades in the ClyA-AS nanopore upon binding of ADP.** A typical ionic current blockades in the presence of 0.1 mM ADP (top), 1 mM ATP (middle) and 1 mM ADP with no Mg²⁺ added (bottom) for type 1 (**A**) and type 2 (**B**) blockades. **C** LID closing frequency - measured as the inverse of dwell times – at increasing ATP concentrations. The lines indicate a linear fit. **D** LID opening frequency at increasing ADP concentrations. The lines indicate a Hill equation fit. **E** Graph showing the percentage closed LID state with increasing concentration of ADP. The lines indicate a Hill equation fit, whereas type I and type II blockades showed a Hill coefficient of 2.3 and 1.7, respectively. On right, the percentage closed LID state with increasing concentration of AMP in addition to 50 μM ADP (*trans*). The measurements were performed in 400 mM KCl, 15 mM Tris, 2 mM MgCl₂ (except when omitted), pH 7.5 at room temperature (22 °C) applying −90 mV (*trans*) and sampling at 50 kHz with a 10 kHz Bessel-low filter, additionally digitally filtered with a 2 kHz Gaussian low-pass filter. Error bars represent the standard deviation of the mean between independent experiments (*N* = 3). Ligands were added to the *trans* chamber and the enzyme to the *cis* chamber.

catalysis[50], were not observed in our nanopore experiments. In this work, we used a higher KCl concentration (400 mM) to achieve a high signal-to-noise ratio. This could influence substrate binding and/or protein dynamics, since Cl⁻ ions compete with phosphates of the substrates in the binding pocket. Although a more physiological ionic concentration could also be used (Supplementary Table 4), molecular modeling showed that under the experimental conditions (400 mM KCl and −80 mV) the concentration of Cl⁻ inside the nanopore is about 100 mM[80], that is similar to the concentration of Cl⁻ in vivo[81]. Alternatively, nanopore confinement might have an influence on protein

dynamics, but our previous work showed that this confinement has a minimal effect on the opening-closing rates for a range of proteins[14,15]. In any case, the environment of the nanopore might be a better proxy for the crowded environment inside a cell compared to the highly diluted solution sampled by other single-molecule techniques. Instead, it is more likely that the high-frequency domain fluctuations are too fast for the sampling rates used in our experiments.

Although $Mg^{2+}$ is required for the closing of the NMP domain, the network of endosteric interactions controlling the hierarchical closing of the domains is independent of $Mg^{2+}$. Structural data of the Ap5A closed conformation shows one $Mg^{2+}$ ion binds to two phosphate groups of the ligand, while no direct interactions between $Mg^{2+}$ and the residues in the LID or NMP domains are observed (Fig. 1B). Domain closing is mediated by the binding of the catalytic residues R156, R123 (LID domain) and R36 (NMP domain)[31] to Ap5A (Fig. 1B). It has been shown that $Mg^{2+}$ positions the substrate optimally for catalysis and organizes the active site through indirect interactions via water molecules[29,35]. This might directly affect the free energy barriers for the opening and closing of both the LID and NMP domains. In light of the data presented here, we argue that cooperative substrate and cofactor binding is likely to have evolved in AK for two reasons. Firstly, to facilitate efficient catalysis through proper substrate orientation. Secondly, to prevent unproductive enzyme closure in the absence of $Mg^{2+}$[81].

This work strongly suggests that AK exists in multiple conformations with different affinity for its ligands. Endoallosteric interactions allow the reshaping of the enzyme by transducing the free energy of ligand/cofactor binding into directional domain motions. This framework can accommodate previously proposed models. As shown in the conformational selection model, multiple shapes exist in AK with different affinity for its ligands. As proposed in the induced-fit mechanism, re-shaping occurs through interactions with ligands. Here, we provide a molecular interpretation, where the collective interplay of the enzyme's soft interactions (e.g., the collective ensemble of all enzymes' interactions) controls the exact hierarchy of large-scale domain motions. Furthermore, given that the ensemble of enzyme interactions contributes to catalysis, this proposed model can explain why enzymes necessitate thermal activation to operate[82] and why mutations distant from the active site can significantly impact catalysis[4-7].

Interestingly, single-molecule analysis also revealed that about 10% of the AK proteins show a compromised ligand binding cooperativity. This subset of enzymes is functional but shows a lower affinity for the second ligand binding (ADP or AMP). The existence of different conformers (i.e., thermally stable enzyme folds with identical sequence but different affinity for ligands) has also been observed for Rabbit Muscle AK[77] and for dihydrofolate reductase[16,17], suggesting it might represent a generic feature in enzyme catalysis. If different molecules will be shown to selectively stabilize one or more of these conformations, this framework could explain how allostery can evolve in enzymes. In AK, both the evolution of allostery to control the binding sequence of ligands and the existence of conformers with varying AMP/ADP/ATP affinities, might enable fine-tuning of the enzyme's activity in response to changing cellular conditions or metabolic demands.

## Methods
### Materials
Ampicillin sodium salt was purchased from Fisher Bio Reagents; Isopropyl β-D-thiogalactopyranoside (≥99.0%, dioxin-free, animal-free), LB medium, 2xYT medium, NaCl (≥99.5%), imidazole (≥99%), KCl (≥99.5%), Tris(2-carboxyethyl)phosphine hydrochloride (≥98.0%), Dodecyl-β-D-maltoside (≥99%), MgCl2 (≥99%), Glucose (≥99,5%) and Lysozyme from Roth. GeneJET Plasmid Miniprep Kit (K0503) GeneJET PCR Purification Kit (K0701), Phire Hot Start II DNA polymerase

(F122S), T4 DNA ligase (EL0011), DNase I (EN00521), *Dpn*I (ER1701), *Nde*I (FD0583) and *Xho*I (ER0691) restriction enzymes were ordered from ThermoFisher Scientific. Ni-NTA agarose from Qiagen; Strep Tactin® Sepharose® and D-desthiobiotin from IBA Lifesciences. diphytanoyl-*sn*-glycero-3-phosphocholine (DPhPC, CAS #207131-40-6), hexadecane 99% (CAS # 544-76-3), hexokinase (CAS #9001-51-8), glucose-6-phosphate-dehydrogenase (900-40-5), NADPH (CAS #2646-71-1), Ap5A (≥95%), ADP (≥95%), ATP (CAS #349-07-08), AMP (CAS #149022-20-8) were obtained from Sigma/Merck. Sequencing was done by Macrogen and DNA primers and gBlock™ from IDT.

### Cloning of AK variants
*AK_wt (AK)* gene containing NdeI and XhoI restriction sites was digested and ligated into a pET22b plasmid. The resulting plasmid was transformed into *E. cloni*® to amplify the DNA and used as template to clone the AK_2+ variants in this paper. Overlap Extension PCR was used to add the two C-terminal lysine residues and the Strep-tag to the *AK_2*+ gene by using the aforementioned *AK* plasmid as well as a DHFR-Strep containing plasmid as templates, resulting in *AK_2 +* . The gene was restricted and ligated into a fresh pET22b plasmid. The point mutations either in the tail of the protein (addition of one Arg, *AK_3 +* ) and at position 156 (*R156A_AK_2 +* ) were introduced in the *AK_2*+ gene via mega primer PCR resulting in the desired full plasmid which could be transformed into *E. cloni*® for DNA amplification. Sequencing confirmed the identity of all produced plasmids of this study.

### DNA and protein sequences
additional residues underlined; the Strep-Tag sequence is in bold.

**ClyA-AS.** ATGACGGGTATCTTTGCGGAACAGACGGTGGAAGTTGTGA
AAAGTGCGATTGAAACGGCTGACGGTGCGCTGGACCTGTATAATAAA
TATCTGGATCAGGTCATCCCGTGGAAAACCTTTGACGAAACGATTAA
AGAACTGAGCCGTTTCAAACAGGAATACAGTCAAGAAGCGTCCGTCC
TAGTGGGCGATATCAAAGTGCTGCTGATGGATTCTCAGGACAAATAT
TTTGAAGCTACCCAAACGGTTTACGAATGGGCGGGTGTGGTTACCCA
GCTGCTGTCCGCATATATTCAGCTGTTCGATGGATACAATGAGAAAA
AAGCGAGCGCGCAGAAAGACATTCTGATCCGCATTCTGGATGACGGC
GTGAAAAAACTGAATGAAGCCCAGAAATCGCTGCTGACCAGCTCTCA
ATCATTTAACAATGCCTCGGGTAAACTGCTGGCACTGGATAGCCAGC
TGACGAACGACTTTTCTGAAAAAAGTTCCTATTACCAGAGCCAAGTCG
ATCGTATTCGTAAAGAAGCCTACGCAGGTGCCGCAGCAGGTATTGTG
GCCGGTCCGTTCGGTCTGATTATCTCATATTCAATTGCTGCGGGCGT
TGTCGAAGGTAAACTGATTCCGGAACTGAACAATCGTCTGAAAACCG
TTCAGAACTTTTTCACCAGTCTGTCTGCTACGGTCAAACAAGCGAATA
AAGATATCGACGCCGCAAAACTGAAACTGGCCACGGAAATCGCTGCG
ATTGGCGAAATCAAAACCGAAACGGAAACCACGCGCTTTTATGTTGA
TTACGATGACCTGATGCTGAGCCTGCTGAAAGGTGCCGCGAAGAAAA
TGATTAATACCTCTAATGAATATCAGCAGCGTCACGGTAGAAAAACCC
TGTTTGAAGTCCCGGATGTGGGCAGCAGCTACCACCATCATCACCAC*

MTGIFAEQTVEVVKSAIETADGALDLYNKYLDQVIPWKTFDETIKELS
RFKQEYSQEASVLVGDIKVLLMDSQDKYFEATQTVYEWAGVVTQLLSAYI
QLFDGYNEKKASAQKDILIRILDDGVKKLNEAQKSLLTSSQSFNNASGKLL
ALDSQLTNDFSEKSSYYQSQVDRIRKEAYAGAAAGIVAGPFGLIISYSIAAG
VVEGKLIPELNNRLKTVQNFFTSLSATVKQANKDIDAAKLKLATEIAAIGEIK
TETETTRFYVDYDDLMLSLLKGAAKKMINTSNEYQQRHGRKTLFEVPDVG
SSYHHHH

**AK_2+.** ATGCGTATCATTCTGCTTGGCGCTCCGGGCGCGGGGAAAGGG
ACTCAGGCTCAGTTCATCATGGAGAAATATGGTATTCCGCAAATCTCC
ACTGGCGATATGCTGCGTGCTGCGGTCAAATCTGGCTCCGAGCTGGG
TAAACAAGCAAAAGACATTATGGATGCTGGCAAACTGGTCACCGACG
AACTGGTGATCGCGCTGGTTAAAGAGCGCATTGCTCAGGAAGACTGC
CGTAATGGTTTCCTGTTGGACGGCTTCCCGCGTACCATTCCGCAGGC
AGACGCGATGAAAGAAGCGGGCATCAATGTTGATTACGTTCTGGAAT
TCGACGTACCGGACGAACTGATCGTTGACCGTATCGTCGGTCGCCGC

GTTCATGCGCCGTCTGGTCGTGTTTATCACGTTAAATTCAATCCGCC
GAAAGTAGAAGGCAAAGACGACGTTACCGGTGAAGAACTGACTACCC
GTAAAGATGATCAGGAAGAGACCGTACGTAAACGTCTGGTTGAATAC
CATCAGATGACAGCACCGCTGATCGGCTACTACTCCAAAGAAGCAGA
AGCGGGGTAATACCAAATACGCGAAAGTTGACGGCACCAAGCCGGTTG
CTGAAGTTCGCGCTGATCTGGAAAAAATCCTCGGCGGC**AAGAAA**TGG
AGCCATCCGCAGTTTGAAAAA*

MRIILLGAPGAGKGTQAFIMEKYGIPQISTGDMLRAAVKSGSELGK
QAKDIMDAGKLVTDELVIALVKERIAQEDCRNGFLLDGFPRTIPQADAMK
EAGINVDYVLEFDVPDELIVDRIVGRRVHAPSGRVYHVKFNPPKVEGKDDV
TGEELTTRKDDQEETVRKRLVEYHQMTAPLIGYYSKEAEAGNTKYAKVDG
TKPVAEVRADLEKILGGKKWSHPQFEK

**AK_3+.** ATGCGTATCATTCTGCTTGGCGCTCCGGGCGCGGGGAAAGG
GACTCAGGCTCAGTTCATCATGGAGAAATATGGTATTCCGCAAATCT
CCACTGGCGATATGCTGCGTGCTGCGGTCAAATCTGGCTCCGAGCTG
GGTAAACAAGCAAAAGACATTATGGATGCTGGCAAACTGGTCACCGA
CGAACTGGTGATCGCGCTGGTTAAAGAGCGCATTGCTCAGGAAGACT
GCCGTAATGGTTTCCTGTTGGACGGCTTCCCGCGTACCATTCCGCAG
GCAGACGCGATGAAAGAAGCGGGCATCAATGTTGATTACGTTCTGGA
ATTCGACGTACCGGACGAACTGATCGTTGACCGTATCGTCGGTCGCC
GCGTTCATGCGCCGTCTGGTCGTGTTTATCACGTTAAATTCAATCCG
CCGAAAGTAGAAGGCAAAGACGACGTTACCGGTGAAGAACTGACTAC
CCGTAAAGATGATCAGGAAGAGACCGTACGTAAACGTCTGGTTGAAT
ACCATCAGATGACAGCACCGCTGATCGGCTACTACTCCAAAGAAGCA
GAAGCGGGTAATACCAAATACGCGAAAGTTGACGGCACCAAGCCGGT
TGCTGAAGTTCGCGCTGATCTGGAAAAAATCCTCGGCGGC**AAGAAAC**
GT**TGGAGCCATCCGCAGTTTGAAAAA***

MRIILLGAPGAGKGTQAFIMEKY-
GIPQISTGDMLRAAVKSGSELGKQAKDIMDAGKLVTDELVIALVKERIAQE
DCRNGFLLDGFPRTIPQADAMKEAGINVDYVLEFDVPDELIVDRIVGRRVH
APSGRVYHVKFNPPKVEGKDDVTGEELTTRKDDQEETVRKRLVEYHQMT
APLIGYYSKEAEAGNTKYAKVDGTKPVAEVRADLE-
KILGGKKR**WSHPQFEK**

**R156A_AK_2+.** ATGCGTATCATTCTGCTTGGCGCTCCGGGCGCGGGGA
AAGGGACTCAGGCTCAGTTCATCATGGAGAAATATGGTATTCCGCAA
ATCTCCACTGGCGATATGCTGCGTGCTGCGGTCAAATCTGGCTCCGA
GCTGGGTAAACAAGCAAAAGACATTATGGATGCTGGCAAACTGGTCA
CCGACGAACTGGTGATCGCGCTGGTTAAAGAGCGCATTGCTCAGGAA
GACTGCCGTAATGGTTTCCTGTTGGACGGCTTCCCGCGTACCATTCC
GCAGGCAGACGCGATGAAAGAAGCGGGCATCAATGTTGATTACGTTC
TGGAATTCGACGTACCGGACGAACTGATCGTTGACCGTATCGTCGGT
CGCCGCGTTCATGCGCCGTCTGGTCGTGTTTATCACGTTAAATTCAA
TCCGCCGAAAGTAGAAGGCAAAGACGACGTTACCGGTGAAGAACTGA
CTACC**GCG**AAAGATGATCAGGAAGAGACCGTACGTAAACGTCTGGTT
GAATACCATCAGATGACAGCACCGCTGATCGGCTACTACTCCAAAGA
AGCAGAAGCGGGTAATACCAAATACGCGAAAGTTGACGGCACCAAGC
CGGTTGCTGAAGTTCGCGCTGATCTGGAAAAAATCCTCGGCGGC**AAG**
**AAA**TGGAGCCATCCGCAGTTTGAAAAA*

MRIILLGAPGAGKGTQAFIMEKYGIPQISTGDMLRAAVKSGSELGK
QAKDIMDAGKLVTDELVIALVKERIAQEDCRNGFLLDGFPRTIPQADAMKE
AGINVDYVLEFDVPDELIVDRIVGRRVHAPSGRVYHVKFNPPKVEGKDDVT
GEELTTA KDDQEETVRKRLVEYHQMTAPLIGYYSKEAEAGNTKYAKVDGT
KPVAEVRADLEKILGGKK**WSHPQFEK**

## Expression and purification of AK proteins

The plasmid containing the *AK* gene was transformed into *E. coli* BL21(DE3) and grown overnight in 20 ml 2-YT medium at 37 °C. This starter culture was transferred into 100 ml fresh 2-YT medium and grown until $OD_{600}$ ≥0.6. After induction with 0.5 mM IPTG, cells were cultivated at 25 °C overnight. The next day, cells were harvested by centrifugation (8000 x $g$, 5 min) and a cell pellet from 100 ml cell culture was resuspended in 10 ml lysis buffer (15 mM Tris, 150 mM NaCl, 10 µg/ml lysozyme, 0.2 U/ml Dnase, 5 mM MgCl2, pH 7.5). After

incubation for 20 min at room temperature and lysis via sonication, solution was centrifuged for 30 min at 6500 x $g$. The cell lysate was transferred to a column packed with 500 µl Strep-Tactin® Sepharose and the flow through was loaded again to obtain maximal loading. The column was washed with 15 mM Tris, 150 mM NaCl, pH 7.5 and the protein was eluted in two steps with 2 ml 5 mM D-Desthiobiotin in 15 mM Tris, 150 mM NaCl, pH 7.5 after incubation for 30 min at 4 °C. The protein was concentrated using Amicon filters and stored at 4 °C.

## Enzymatic bulk assay

Adenylate kinase activity was determined in a time-dependent coupled enzyme assay using ADP as substrate and the enzymes hexokinase and glucose-6-phosphate-dehydrogenase to finally produce NADPH, which can be monitored spectroscopically. For this, 10 mM glucose, 0.5 mM NADPH, 0.006 U/µl hexokinin, 0.003 U/µl glucose-6-phosphate-dehydrogenase and 2 mM ADP were added to 200 µl of reaction buffer (15 mM Tris, 400 mM KCl, 2 mM MgCl$_2$). After addition of 30 nM AK, the reaction was monitored at an absorbance of 340 nm for 180 s.

## Expression and purification ClyA-AS protein

*E. cloni*® EXPRESS BL21 (DE3) cells were transformed with the pT7-SC1 plasmid containing the *ClyA-AS* gene. ClyA-AS contains eight mutations relative to the *S. Typhi* ClyA-WT: C87A, L99Q, E103G, F166Y, I203V, C285S, K294R and H307Y (the H307Y mutation is in the C-terminal hexahistidine-tag added for purification)[70]. Transformants were selected after overnight growth at 37 °C on LB agar plates supplemented with 100 µg/mL ampicillin. The resulting colonies were grown at 37 °C (200 rpm shaking) in 2xYT medium supplemented with 100 µg/mL ampicillin until the O.D. at 600 nm was ~0.8. ClyA-AS expression was then induced by addition of 0.5 mM IPTG, and the temperature was switched to 25 °C for overnight growth (200 rpm shaking). The next day the bacteria were harvested by centrifugation at 6000 x $g$ for 25 min at 4 °C and the pellets were stored at −80 °C.

Pellets containing monomeric ClyA-AS arising from 50 mL culture were thawed and resuspended in 20 mL of wash buffer (10 mM imidazole, 150 mM NaCl, 15 mM Tris-HCl pH 7.5), supplemented with 1 mM MgCl2 and 0.2 units/mL of DNaseI. After lysis of the bacteria by probe sonication, the crude lysates were clarified by centrifugation at 6000 x $g$ for 20 min at 4 °C and the supernatant was mixed with 200 µL of Ni-NTA resin equilibrated in wash buffer. After 1 hr, the resin was loaded into a column (Micro Bio Spin, Bio-Rad) and washed with ~5 mL of the wash buffer. ClyA-AS was eluted with approximately ~0.5 mL of wash buffer containing 300 mM imidazole. ClyA-AS monomers were stored at 4 °C until further use.

ClyA-AS monomers were oligomerized by addition of 0.5% β-dodecylmaltoside and incubation at 37 °C for 30 min. ClyA-AS oligomers were separated from monomers by blue native polyacrylamide gel electrophoresis (BN-PAGE, Bio-rad) using 4–20% polyacrylamide gels. The bands corresponding to Type I ClyA-AS were excised from the gel and placed in 150 mM NaCl, 15 mM Tris-HCl pH 7.5, supplemented with 0.2% DDM and 10 mM EDTA to allow diffusion of the proteins out of the gel. The concentration of the proteins was measured using the Bradford assay.

## Electrical recordings in planar lipid bilayers

The setup consists of two buffer-filled chambers (*cis* and *trans*) separated by a 20 µm teflon film (Goodfellow Cambridge Limited) containing a central aperture with a diameter of ~100 µm. The aperture was pre-treated with hexadecane (5% (v/v) in pentane) followed by addition of 500 µl of buffer (400 mM KCl, 15 mM Tris, pH 7.5) to both chambers. For lipid bilayer formation, two droplets of 1,2-diphytanoyl-sn-glycero-3-phosphocholine (DPhPC) were added to both chambers and a negative potential was applied to the *trans* side of the setup. Lowering and raising the buffer level facilitates the lipid bilayer which spontaneously forms. For a single nanopore insertion, a pipette tip was

dipped into a solution of pre-oligomerized ClyA-AS and dipped afterwards into the buffer of the *cis* compartment.

For electrical recordings of adenylate kinase -100 nM protein was added to the *cis* compartment while the ligands (ATP, ADP, AMP, Ap5A) were added to the *trans* side. For measurements without $Mg^{2+}$, 100 μM EDTA was added to the buffer to prevent contamination.

## Data recording and processing

Electrophysiological data were collected applying negative potential (*trans*) using an Axopatch 200B patch clamp amplifier (Axon Instruments) connected to a DigiData 1440 A/D converter (Axon Instruments). Data was sampled at 50 kHz and a low-pass Bessel filter of 10 kHz. The traces were additionally digitally filtered with a 2 kHz low-pass Gaussian filter. Data was recorded using Clampex 10.7 software (Molecular Devices) and analyzed by Clampfit 10.7 software (Molecular Devices).

## Electrophysiological data analysis

The residual current was determined by calculating $I_{RES} = (I_B/I_O)*100$, whereas $I_O$ reflects the open pore current and $I_B$ the current during a protein blockade. $I_B$ can therefore be calculated for the average protein blockade (e.g. medium blockades in this work) or for individual current levels within the protein blockade (e.g. M2 in this work). Mean $I_O$ and $I_B$ were determined from Gaussian fits to all-point histograms (bin width 0.1 pA) of at least 20 current blockades from at least three single channels using Clampfit 10.7 (Molecular Devices).

Blockade types (deep, medium, shallow) were counted by selecting them via single-channel search function ignoring events shorter than 1 ms. Occurrence of the blockade types (%) was calculated as ratio of the blockade counts from the sum of all blockades within the trace. Mean dwell times for each blockade type were determined by exponential fitting of all dwell times of one blockade type detected via single-channel search in three single channels binned with a width close to the expected dwell time.

The occurrence of a certain level (M, M1, M2 or M3 in %) within the medium protein blockade was determined by calculation of all-point histograms (bin width 0.1 pA) of at least 15 current blockades of at least three single channels. Within the histogram, peaks representing the individual levels were identified. The counts corresponding to the maximum of the peak of interest was divided by the sum of the counts of the maxima of all relevant peaks within the histogram. Occurrence of M2 (%) was plotted against the ligand concentration and fitted with a Hill equation using Origin (OriginLab Corporation). From this fitting, $K_D$ and hill coefficient could be determined.

Event frequencies (1/tau, e.g. opening frequency or closing frequency) were calculated from event dwell times (tau) which were determined by single channel search function ignoring events shorter than 0.1 ms. The mean event dwell times were determined by exponential fitting of at least 1000 events per concentration and of at least three single channels that were binned with a width close to the expected mean dwell time. Closing frequencies were plotted against the ligand concentration and fitted with a linear fit while opening frequencies were fitted with a Hill function or a linear function using Origin (OriginLab Corporation).

Individual transitions between the different states were determined by single channel search (at least 1000 events, events shorted than 0.1 ms were ignored) within medium blockades. Afterwards, the events were sorted by the event which comes next resulting in six data sets of transitions: M→M2 and M→M3; M2→M and M2→M3; M3→M2 and M3→M2. For each level, the occurrence of transition (%) was determined by dividing the number of one transition type (e.g. M→M3) by the sum of all possible transitions from this level (e.g. M→M3 + M→M2). This calculation was done for at least three single channels.

For the determination of transitions from M2→M3 (%) at different $Mg^{2+}$ concentrations, M2 events were detected by single channel

search and sorted by the level which came next. The occurrence of transitions from M2→M3 (%) was than calculated as a fraction of the sum of all transitions starting from M2 (M2→M3 + M2→M). M2→M3 transitions (%) at different $Mg^{2+}$ concentrations were fitted to a one-site specific binding curve using Prism 5 (GraphPad).

## Hidden Markov model construction and parameter estimation

**Data selection and preparation.** The data sample used for inference consisted of time series of current measurements corresponding to different enzymes trapped in nanopores. The time series were acquired at varying ADP and ATP/AMP concentrations ranging from 0.01 to 1.00 mM, with two to four replicates per experimental condition and MgCl2 concentrations fixed to 2 mM. The time series lengths range from 500 to 10,000 ms, and in total, the data consisted of 23 time series for varying ADP concentrations, 21 time series for varying ATP concentrations, and 19 time series for maximal ATP and varying AMP concentrations.

From this set of time series, usable segments were selected to exclude open pore current levels (around −400 pA) and fragments with mean levels substantially higher than −180 pA (−160 pA and above). This resulted in 58,301 ms of data spread over 39 segments with varying ADP concentrations, 101,040 ms of data spread over 51 segments with varying ATP concentrations, and 84,545 ms of data spread over 35 segments with maximal ATP and varying AMP concentrations.

To estimate model parameters, we selected a random segment for each experimental condition from a representative set of concentrations. For the ADP model, this set consisted of 0.01, 0.02, 0.05, 0.10, 0.20, 0.50, and 1.00 mM of ADP, and for the ATP/AMP model, they were 0.01/0.00, 0.10/0.00, 1.00/0.00, 1.00/0.01, 1.00/0.10, and 1.00/1.00 mM of ATP/AMP. Each random segment was truncated on the right to the least of either the length of the shortest segment or 1000 ms.

**Hidden Markov model and likelihood.** We assumed the data were generated from the (continuous time) hidden Markov model[83] illustrated in Fig. 3, with states denoted as M, M*, M**, M1, M1*, M1**, M2*, M2**, and M3**. M, M1, M2, or M3 indicate the mean emission level and each asterisk represents a bound ligand (in the case of the ATP/AMP model, one asterisk corresponds to ATP and two to ATP + AMP). We included an additional M0 state to capture sporadic transitions to a mean level around −180 pA, which can be visited from M, M*, M1, and M1*. Although uncommon, visits to this anomalous level were too frequent to eliminate from the time series since the remaining continuous segments would become too short. Putting everything together, we arrived at the following matrix (*Q*-matrix) of transition rates for the continuous-time Markov chain of hidden states:

$$
Q := \begin{pmatrix}
\sim & c_1 & c_2 & c_3 & c_4 & & & & & \\
c_{-1} & \sim & k_1|ATP| & b_{-1} & & & & & & \\
c_{-2} & k_{-1} & \sim & & b_{-1} & & & & & \\
c_{-3} & b_1 & & \sim & k_1|ATP| & & & & & \\
c_{-4} & & b_1 & k_{-1} & \sim & r_1 & & & & \\
& & & & r_{-1} & \sim & k_2|AMP| & & & \\
& & & & & k_{-2} & \sim & r_{-3} & r_{-4} & b_2 \\
& & & & & & r_3 & \sim & & \\
& & & & & & r_4 & & \sim & \\
& & & & & & b_{-2} & & & \sim
\end{pmatrix}
\begin{matrix}
M0 \\ M1 \\ M1* \\ M \\ M* \\ M2* \\ M2** \\ M** \\ M1** \\ M3**
\end{matrix}
$$

(1)

with |ATP| and |AMP| expressed in mM. A similar model for ADP-dependent conformational dynamics is presented in the Supplement (Supplementary Figs. 21, 22). The constants $c_{\pm 1}, \ldots, c_{\pm 4}$, $k_{\pm 1}$, $k_{\pm 2}$, $b_{\pm 1}$, $b_{\pm 2}$, $r_{\pm 1}, \ldots, r_{\pm 4}$ denote the (positive) forward and backward transition rates and the tildes $\sim$ represent the negative sums of the off-diagonal elements for each row.

To facilitate the likelihood calculation, the continuous-time Markov chain defined above was converted into a discrete-time one with the same state space and a time step $\Delta t = 0.2$ ms equal to the data sampling rate, ensuring that no fast transitions were missed and that the discrete-time model resembled the continuous-time one on the timescale defined by the sampling interval of the measurements. The transition probability matrix of the discrete-time chain is defined as

$$\boldsymbol{\Gamma} := \exp(\Delta t \cdot \boldsymbol{Q}) \tag{2}$$

The measurements were assumed to represent emissions from a realization of the underlying discrete-time Markov chain, with normal emission distributions conditional on the hidden state. We assumed that LID and NMP movements give rise to a distinct mean level, but ligand binding and unbinding is undetectable. As such, we constrained hidden states with the same LID and NMP arrangements to have equally distributed emissions. Supplementary Table 13 lists the approximate mean emission per group of states, but it should be noted that these values vary slightly across different time series in the data and were therefore optimized within a short interval for each data segment. All standard deviations of the emission distributions varied between 7 and 10, in line with the estimated standard deviations of the measurements. Given the assumptions above, the probability density of measuring current level $y$ at time $t$ is

$$p(y(t)) := \sum_{m \in \{M0, M1, M1^*, \ldots, M3^{**}\}} \phi(y(t); \mu(m(t)), \sigma(m(t))) \cdot p(m(t)), \tag{3}$$

where $\mu(m)$, $\sigma(m)$ denote the mean and standard deviation of the emission distribution of state $m$, $\phi(y; \mu, \sigma)$ denotes the probability density function of the normal distribution with parameters $\mu, \sigma$ at $y$, and $p(m(t))$ is the probability of finding the underlying Markov chain in state $m$ at time $t$. The joint likelihood of the vector of rates

$$\boldsymbol{k} = (c_{\pm 1}, \ldots, c_{\pm 4}, k_{\pm 1}, k_{\pm 2}, b_{\pm 1}, b_{\pm 2}, r_{\pm 1}, \ldots, r_{\pm 4}), \tag{4}$$

The vector of mean emissions $\boldsymbol{\mu}$, and the vector of emission standard deviations $\boldsymbol{\sigma}$ given a time series of current level measurements $\boldsymbol{y} := (y_1, y_2, \ldots, y_T)$ is then given by[83]

$$L(\boldsymbol{k}, \boldsymbol{\mu}, \boldsymbol{\sigma}; \boldsymbol{y}) = \boldsymbol{\delta}' \left( \prod_{t = 1, 2, \ldots, T} \boldsymbol{B}_t \right) \boldsymbol{1}, \tag{5}$$

where $\boldsymbol{\delta}'$ denotes the (row) vector containing the stationary state distribution, $\boldsymbol{1}$ is the 10-dimensional column vector of ones, and

$$\boldsymbol{B}_t := \boldsymbol{\Gamma} \, \text{diag}\big(\phi(y_t; \mu(m), \sigma(m))\big)_{m \in \{M0, M1, M1^*, \ldots, M3^{**}\}} \tag{6}$$

When evaluating the likelihood $L(\boldsymbol{k}, \boldsymbol{\mu}, \boldsymbol{\sigma}; \boldsymbol{y})$, intermediate products $\boldsymbol{\delta}' \prod_{t = 1, \ldots, \tau} \boldsymbol{B}_\tau$ were normalized to avoid underflow[83]. To fit current level measurements from multiple experimental setups simultaneously, we computed the likelihood of the system parameters given multiple time series $\boldsymbol{y}_1, \boldsymbol{y}_2, \ldots$ Given that individual experiments were independent of each other, the resulting likelihood is simply the product of the individual likelihoods, i.e.,

$$L(\boldsymbol{k}, \boldsymbol{\mu}, \boldsymbol{\sigma}; \boldsymbol{y}_1, \boldsymbol{y}_2, \ldots) = \prod_{i = 1, 2, \ldots} L(\boldsymbol{k}, \boldsymbol{\mu}, \boldsymbol{\sigma}; \boldsymbol{y}_i) \tag{7}$$

where $i$ runs over the different experiments.

*Estimation* For likelihood maximization, we used the interior point method implemented by fmincon in MATLAB with step tolerance 1E-3 and a maximum of 200 iterations. The model parameters were bound between 1E-1 and 1E6, except for a few specific adjustments made (1) to constrain transition rates for which rough bounds could be directly inferred from the data, as they were not affected by different hidden states and (2) to prevent the optimization procedure from getting stuck in local optima. The MATLAB script used for parameter estimation are provided on GitHub[84]. Each optimization was multi-started from six random initial points, log-uniformly sampled within the bounds.

**Prediction of LID closing and opening rates.** In our model, the experimentally observed LID closing/opening rates correspond to aggregate rates to/from the group of LID-closed states M2*, M2**, M3** respectively. To estimate these rates from the model for each ligand concentration, the intensity matrix $\boldsymbol{Q}$ and the transition matrix $\boldsymbol{\Gamma}$ were computed, and a sample path of 20,000 ms was simulated from the corresponding Markov chain using standard techniques. The dwell times in the groups of LID-closed and LID-open states were then recorded, and the inverse dwell times for the group of open (closed) states were designated as the LID closing (opening) rates.

### Reporting summary
Further information on research design is available in the Nature Portfolio Reporting Summary linked to this article.

## Data availability
Source data is deposited in the Zenodo database under the accession DOI code: https://doi.org/10.5281/zenodo.14047871. PDB codes of previously published structures used in this study are 4AKE and 1AKE.

## Code availability
"The Nanopore HMM software package is available at: https://github.com/yulanvanoppen/nanopore-HMM/tree/v1.0.0[84] and also deposited in the ZENODO database under the accession DOI code: https://doi.org/10.5281/zenodo.13911924.

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

## Acknowledgements

We thank Stefania Digiovanni, Lisa Jacob and Guangnan Shi for preliminary studies for this work. S.Z. was funded by an ERC consolidator grant number: 726151. N.G. and MvdN were funded by a NOW-VICI grant: 192.068. Y.v.O. and A.M.-A. were supported by the Dutch Research Council (Nederlandse Organisatie voor Wetenschappelijk Onderzoek; NWO) through an NWO-VIDI grant to A.M.-A. (project number 016.Vidi.189.116).

## Author contributions

N.G., S.Z. and G.M. designed the study. N.G., S.Z. and M.vdN. performed all experiments, analyzed the data. Y.vO. and A.M.A. constructed and analyzed the mathematical model. N.G. and G.M. wrote the paper. G.M. supervised the project. All authors verified the manuscript.

## Competing interests

G.M. is founder, director, and shareholder in Portal Biotech Limited, a nanopore company. This work was not supported by Portal Biotech Limited. The remaining authors declare no competing interests'.
