## [Transparent Peer Review file · Nature Communications]

Allostery can convert binding free energies into concerted domain motions in enzymes

Corresponding Author: Professor Giovanni Maglia

Version 0:

Reviewer comments:

Reviewer #1

(Remarks to the Author)

In the manuscript by Maglia and co-workers, enzymology and structural dynamics of the enzyme adenylate kinase (AK) is approached with single molecule nanopore spectrometry. I am not an expert in the nanopore technique but well familiarized with the enzyme, therefore my report will primarily be focused on the enzyme but with some comments on the spectroscopic approach. The main conclusion in the manuscript is a sequential binding mechanism of the substrates ATP and AMP where ATP is found to bind prior to AMP. Also the experiments allows the observation of transition between multiple structural states that collectively contributes to impressive details of the energetic landscape of AK. Overall, the manuscript adds significant knowledge to the field of enzyme dynamics and as such is suitable for publication in nature communications. Despite the interesting new data presented there are some clarifications and comparisons with previous findings that are necessary. These issues are described under "major points" below, and for some smaller issues I list them under "minor points"

Major points:

- 1) The data is acquired with conditions of 400 mM KCl, this poses a potential problem since the interaction with phosphates is of electrostatic nature. I suggest that the authors present activity data with much less salt in the assay and compare these values to the 400 mM KCl case..
- 2) The activity data is best presented as k_{cat} and K_m , and should be benchmarked to previous studies.
- 3) Can the authors strengthen the basis for assignment of the various states, i.e M through M4. Does the degree of blockage scale with the accessible surface area of the states?
- 4) In previous single molecule FRET, NMR and crystallography studies the fully closed state (M3 and M4 in the current manuscript) is the dominant state with saturating amounts of ADP or ATP/AMP. In the current manuscript the M3 state in Fig 3E, G and Fig 4 is only minutely populated. The authors should explain potential reasons for these discrepancies.
- 5) Equilibrium constants for the various dynamic transition should be computed from the microscopic rate constants and also compared to previous findings.
- 6) The rate constants for the transitions should be compared with published values and similarities and discrepancies should be discussed.
- 7) I think that a simplified figure that summarizes the finding of the allosteric effect can be added as an additional and final image linked to the discussion/conclusion section.

Minor points:

- 1) Figure 1C would benefit from a schematic description of both Typ-a and Typ-b blockade. I believe only type-a is shown.
- 2) In the third sentence of the introduction I think that "...with identical structures" should be replaced with "...with identical active-site structures".
- 3) In figure 1C the equilibrium between open and blocked nanopore can be quantified from the dwell times. Is this equilibrium far shifted towards the blocked state in the experiments? This can potentially be addressed from the equilibrium constant and the enzyme concentration used in experiments
- 4) AMP has previously been proposed to inhibit AK by binding to the ATP site. In the current manuscript AMP inhibition is explained with an allosteric model. A discussion and comparison of both these models would be interesting.
- 5) The biological relevance of the findings are described a bit hand-waiving as a means to prevent water mediated hydrolysis of ATP and ADP. Are there other potential fitness benefits? Can the cellular concentrations of AMP and ATP be a clue?

Reviewer #2

(Remarks to the Author)

The manuscript describes a study using a ClyA nanopore to confine adenylate kinase (AK) and investigate ligand-induced conformational changes of AK, manifested as ionic current fluctuations. Various ligands, such as the inhibitor Ap5A, ATP, AMP, and ADP, were tested. The addition of each ligand, except for AMP, triggered current signal changes which were attributed to the ATP lid closed and ATP lid closed + AMP lid closed states. The data convincingly show that ATP binding is enhanced in the presence of AMP, demonstrating that AMP binding allosterically affects ATP lid closure. The experiments were well-designed, and the data analysis was carefully executed. The study provides concrete evidence of the allosteric effect by which the two ligand bindings regulate the lid domain movements. Directional lid domain closures, i.e., ATP lid closure preceding AMP lid closure, are supported by the observation that AMP alone did not induce any noticeable current signals. The conclusions are largely supported by the experiments. However, the manuscript can be improved in clarity, accuracy, and readability by addressing the following issues listed below:

1. The Catalytic Kinetic Model: In the proposed model, the closing of the NMP domain is suggested to be the rate-limiting step of the catalytic cycle. Does this observation agree with previous studies? Please discuss how the present study supports or argues against previous findings.
2. Figure 2c: The M1 state was interpreted as the AMP lid closed state. Please provide the reasoning for this interpretation.
3. Figure 2E and 2G: In Figure 2E, the M2 signal is assigned to the ATP loaded + lid closed state, while in Figure 2G, M2 is shown as ATP loaded + AMP loaded + lid closed. This is confusing. The M2 state in Figure 2E should also be described as ATP loaded + AMP loaded + lid closed for consistency.

The writing of the manuscript should be improved. It contains many words and phrases that are too vague to understand. Examples are listed below:

5. In the abstract: It states, "Here, we investigate the catalytic cycle of adenylate kinase (AK), an enzyme that catalyzes the interconversion of the various adenosine phosphates (ATP, ADP, and AMP)." This statement is incorrect. The manuscript does not investigate the catalytic cycle, which would focus on the transfer of the gamma-phosphate from ATP to AMP.
6. In the abstract: The authors state, "...that allosteric interactions enable converting ligands and cofactor binding energies into directional conformational changes of the two catalytic domains of AK." In biochemistry, allosteric interactions usually refer to interactions between the multi-subunits of oligomeric proteins. In AK, the binding of two ligands at different sites allosterically affects the lid and NMP's closure. The ligand interaction with AK at its active sites should not be called allosteric interactions.
7. The first paragraph of the introduction: The content is irrelevant to the present study. This paragraph describes transition state theory and its application and limitations in explaining the enzyme's mechanism. However, the experiments in the manuscript do not study any aspect of the transition state of AK.
8. On page 10: The authors wrote, "These observations suggest the existence of an endosteric effect whereas: the initial binding of AMP inhibits the binding of ATP, or the initial binding of AMP forms a less active conformation, as previously suggested, or the ATP-induced closing of the LID domain induces the formation of a higher-affinity binding site for AMP." How is the endosteric effect defined? What is the difference between this "endosteric" effect and the common biochemistry concept of "cooperativity"? Based on the author's description of the endosteric effect as AMP binding inhibiting ATP binding, it is just a classic negative cooperativity behavior. It seems that the authors invented a new term to replace a common term in the textbook, which decreases the readability of the manuscript.
9. On page 16: The authors mention, "where the collective interplay of the enzyme's soft interactions controls..." What do "soft" interactions mean? The term "soft" is not defined or explained in the manuscript.

Reviewer #3

(Remarks to the Author)

I am commenting on the manuscript from the perspective of an expert in the study of conformational transitions (including allosteric communication) in various proteins, but I am not familiar with the relevance of the protein studied here - adenylate kinase (AK). However, I have searched the literature, and from the publications I have found, I believe that the authors have indeed addressed an open and relevant question and gained new insights. In particular, by using single-molecule nanopore spectrometry, they were able to show the sequence of ligand binding events (ATP, ADP and AMP) and how some of these events are controlled by allostery.

The experiments seem to have been performed with care. However, some of the analysis lacks diligence. The determination of the hidden Markov model seems to be in order, however the outcome of it is not surprising.

The manuscript is mostly clearly written.

In principle, the presented data could become publishable in Nature Communications. However, not all of the conclusions drawn are supported by the data. A number of issues need to be addressed before I can recommend this manuscript for publication. I list them in the order in which they appear in the manuscript, some of them are minor, others are important:

- 1) The methods used, i.e., nanopore spectrometry and hidden Markov modeling should be mentioned in the Abstract.
 - 2) Fig. 1A needs to be shown larger. ATP, AMP and Mg²⁺ need to be shown in brighter colors so that readers who are not that familiar with the system obtain a better understanding of the molecular details right at start of the manuscript.
 - 3) Following on from my comment 2), I found parts of the introduction not very informative. The first paragraph of the Introduction is very general and does not really have anything to do with the problem under investigation, while almost no molecular details about AK are given. Along with a more detailed look at Fig. 1A, I would like to know more about the key amino acids for ATP/ADP/AMP and Mg²⁺ binding, as this is important for understanding the binding events and their sequence. I will come back to this comment below in comment 10.
 - 4) On p. 4, the authors mention Type-a, Type-b and Type-c orientations of how AK enters the nanopore. They say that they focus on Type-a blockades. On p. 13, they talk about type II blockades. Are these then the Type-b blockades mentioned earlier? For these, they give a probability of 12±6%, but the probabilities of all other blockades are not given. And do the 12% mentioned on p. 13 correspond to the 10% mentioned on p. 17? This has to be reported with greater care and all details given. And what is with the Type-c / type III blockades?
 - 5) p. 7, 1st sentence: I guess that "that AMP increased the closed probability in the presence of AMP" should read "that AMP increased the closed probability in the presence of ATP". Please check.
 - 6) p. 7: "When AMP was sampled in the presence of ATP" sounds strange. I think that "added" instead of "sampled" would be a better word here. Further below on that page, a similar such occurrence appeared, where "present" would be a better description of the situation.
 - 7) Based on Figs. S9 and S10 the authors concluded that AMP did not induce the closing of the NMP domain. However, when I look at Fig. S9 I count 15 M1 events within 14 ms (and even 4 M2 events) and in Fig. S10 (no Mg²⁺) there are 7 M1 and 4 M2 events within 5 ms, whereas in Fig. 2A there are only 2 M1 events in > 600 ms. To me this seems that the addition of AMP greatly increased the closing of the NMP domain (and sometimes even causes the LID domain to close). Please explain why you interpret Figs. S9 and S10 quite differently than I do and conclude that M1 cannot exist when only AMP is added.
 - 8) p. 10, 1st paragraph: Is the colon before "whereas" put on purpose there? And are the half-sentences after the colon thought as statements or possibilities? For the statement "the initial binding of AMP inhibits the binding of ATP" I could not find any data in the manuscript or SI. Please add! Without clear demonstration of this, the conclusions drawn may partly crumble.
 - 9) p. 10 / Fig. 3: The reader has to guess what the star(s) at states M_x mean. My interpretation is that these are the hidden states where ATP/AMP has bound but the conformational transitions have not occurred yet. However, when looking at Fig. 3, this does not apply to M₂^{*} and M₃^{**} as these are closed states which can be detected by nanopore spectrometry. And why have states M₂ and M₃ vanished altogether in Fig. 3? What are the numbers at the arrows: times, rates, concentrations? Units would be great and a better notation - it took me a while to understand what the "E" meant.
- A side comment is in place: The authors assume that AxP binding precedes the conformational changes; yet this is not shown here and might not be the case; these could also be concerted AxP binding events!
- 10) The Conclusions would greatly benefit from adding a discussion / interpretation of the findings on a more detailed structural level (amino-acid level) to rationalize them (see also my comment 3).
 - 11) p. 27: Please explained the meaning of "y" and what "pdf" stands for.

Version 1:

Reviewer comments:

Reviewer #1

(Remarks to the Author)

The authors has responded to the issues raised by me in a satisfactory manner. The manuscript is now suitable for publication. /Magnus Wolf-Watz

Reviewer #2

(Remarks to the Author)

The authors have addressed the issues raised in the previous review cycle. I recommend the manuscript be accepted for publication.

Reviewer #3

(Remarks to the Author)

The authors carefully addressed all comments, which improved the manuscript. I recommend this revised manuscript for publication on Nature Communications.

We thank the reviewers for carefully reviewing our work. We are glad they have found our work interesting and worth publishing in Nature Communications. We also thank the reviewers for the many insightful comments. We believe the manuscript now provides a strengthened interpretation of the data and it contains a more complete overview of AK catalysis.

Below is our point-by-point answer to the reviewers' comments. In blue is our answer, in italics the extracted text from the manuscript.

REVIEWER COMMENTS

Reviewer #1 (Remarks to the Author):

In the manuscript by Maglia and co-workers, enzymology and structural dynamics of the enzyme adenylate kinase (AK) is approached with single molecule nanopore spectrometry. I am not an expert in the nanopore technique but well familiarized with the enzyme, therefore my report will primarily be focused on the enzyme but with some comments on the spectroscopic approach. The main conclusion in the manuscript is a sequential binding mechanism of the substrates ATP and AMP where ATP is found to bind prior to AMP. Also the experiments allow the observation of transition between multiple structural states that collectively contribute to impressive details of the energetic landscape of AK. Overall, the manuscript adds significant knowledge to the field of enzyme dynamics and as such is suitable for publication in nature communications. Despite the interesting new data presented there are some clarifications and comparisons with previous findings that are necessary. These issues are described under "major points" below, and for some smaller issues I list them under "minor points"

Major points:

1) The data is acquired with conditions of 400 mM KCl, this poses a potential problem since the interaction with phosphates is of electrostatic nature. I suggest that the authors present activity data with much less salt in the assay and compare these values to the 400 mM KCl case.

We have performed the experiments asked by the reviewer. We found a slight dependence of the on and off rate for the binding of ATP to the enzyme. As mentioned by the reviewer, this is perhaps expected, as Cl⁻ are expected to compete with ATP/ADP/AMP in the binding pocket of the enzyme. We have added the data in table S4 and commented in the main text in the discussion about it.

However, it should also be noted that the exact ionic strength composition inside the nanopore is not entirely known. The nanopore is highly cation selective, and inside the nanopore, Cl⁻ ions are depleted (R1). The applied potential has an accentuating effect. (R1) Based on our previous modeling, under 400 mM KCl and -80 mV, the concentration of Cl⁻ inside the nanopore is about 100 mM (Figure 3a of reference R1). We expect, therefore, that in 400 mM KCl, the concentration of Cl⁻ is actually similar to that *in vivo*.

We have mentioned this point in the manuscript:

Notably, recent experimental and simulations findings^{51,58,85-88} have reported LID domain motions on the order of microseconds for both the apo- and liganded enzyme (a rate of 22,000 s⁻¹,⁵¹). These large-scale LID-domain conformational dynamics, which might be required to position the substrates for their optimal orientation during catalysis,⁵¹ were not observed in our nanopore experiments. In this work, we used a higher KCl concentration (400 mM) to achieve a high signal-to-noise ratio. This could influence substrate binding and/or protein dynamics, since Cl⁻ ions compete with phosphates of the substrates in the binding pocket. Although a more physiological ionic concentration could also be used (Table S4), molecular modeling showed that under the experimental conditions (400 mM KCl and -80 mV) the concentration of Cl⁻ inside the nanopore

is about 100 mM⁸⁹, that is similar to the concentration of Cl⁻ in vivo. Alternatively, nanopore confinement might have an influence on protein dynamics, but our previous work showed that this confinement has a minimal effect on the opening-closing rates for a range of proteins.^{90,91} In any case, the environment of the nanopore might be a better proxy for the crowded environment inside a cell compared to the highly diluted solution sampled by other single-molecule techniques. Instead, it is more likely that the high-frequency domain fluctuations are too fast for the sampling rates used in our experiments.

Reference

R1) Willems, K. et al. Accurate modeling of a biological nanopore with an extended continuum framework. *Nanoscale* 12, 16775–16795 (2020).

2) The activity data is best presented as k_{cat} and K_M , and should be benchmarked to previous studies.

We have measured the k_{cat} and K_M for AK and they compare reasonably well with published data (Table S6).

For nanopore data, however, we can only measure the conformational changes associated with the binding of the substrates to the active site. These data cannot be easily connected to the k_{cat} and K_M values resulting from the catalytic activity of the enzyme, because not each binding event necessarily induces domain closing.

3) Can the authors strengthen the basis for assignment of the various states, i.e M through M4. Does the degree of blockage scale with the accessible surface area of the states?

We thank the reviewer for this observation. Yes. In nanopore analysis, the more compacted structure blocks more current. This is because the protein can penetrate deeper inside the nanopore.

The assignment of the four M states is compatible with a physical model for protein-nanopore interactions, which proposes that when the protein's structure becomes more compacted, it penetrates deeper inside the nanopore, resulting in a higher current block. A full kinetic scheme for the binding of ATP and AMP is shown in Figure 3.

4) In previous single molecule FRET, NMR and crystallography studies the fully closed state (M3 and M4 in the current manuscript) is the dominant state with saturating amounts of ADP or ATP/AMP. In the current manuscript the M3 state in Fig 3E, G and Fig 4 is only minutely populated. The authors should explain potential reasons for these discrepancies.

Our data show indeed that the M2 state is the dominant state in the presence of ligands. This does not contradict the FRET experiment, which can only monitor the motions of the LID domain. Likewise, NMR studies (<https://doi.org/10.1038/nsmb821>) reported the global dynamics of the enzyme and did not distinguish between two substrate lids.

We have added lines in the Results and Discussion sections about these two works and their connection with M2

Results: M2 most likely represents the LID domain motions measured by single-molecule FRET⁴⁵ or AFM.⁷² Notably, the fast domain motions induced by ATP, as observed by FRET, are likely too fast to be observed here.

Discussion: In this work, we studied the conformational changes of AK using nanopore technology. This approach allows sampling the global dynamics of the enzyme with high-microsecond resolution for extensive time. We observed three independent LID-NMP domain motions in AK. By comparison, FRET could only measure the motions of the LID domain,⁵¹ while AFM⁷⁹ and NMR⁸⁴ measured the combination of both domains.⁸¹

5) Equilibrium constants for the various dynamic transition should be computed from the microscopic rate constants and also compared to previous findings.

6) The rate constants for the transitions should be compared with published values and similarities and discrepancies should be discussed.

Points 5 and 6 are argued together.

We added all the rate constant values from the literature in Table S6. We have discussed them in the manuscript. However, it is not possible to relate these ensemble values to the data in Figure 3. This is because the equilibrium constants are likely to include multiple states observed in the single-molecule transitions. Furthermore, our single-molecule data revealed a rich set of conformational transitions, which are not visible in previous single-molecule FRET work.

We have added available literature information on the rate constants of AdK in Table S6 and we have discussed them in the manuscript. However, relating these ensemble values to the microscopic rate constants in our model is not possible. This is because transition rates reported in the literature often correspond to transitions from multiple states into multiple states in our model, for which a single rate cannot be computed. Furthermore, our single-molecule data revealed a set of conformational transitions which were not visible in previous single-molecule FRET work (from which literature rates have been derived). Finally, in our experiments the catalytic step cannot be discerned from unproductive domain motions.

The ability of to measure both LID and NMP domain motions independently revealed a precise hierarchy of domain closure during AK catalysis, controlled by endosteric interactions. The initial binding of ATP or ADP induces the closing of the LID domain at a rate of 1000 s^{-1} , which matches the domain closing rate that was measured by NMR for the combined NMP+LID domain motions ($1374 \pm 110\text{ s}^{-1}$).⁸⁴ The LID domain may quickly reopen with a similar rate (1000 s^{-1} , Figure 3). However, if another adenosine phosphate subsequently binds, it induces an endosteric effect that retards the opening of the LID domain and allows the additional closing of the NMP domain (Figure 2J). The closing of the NMP domain has the slowest observed rate (200 s^{-1}), which matches the catalytic turnover rate of the enzyme ($263 \pm 30\text{ s}^{-1}$).⁸⁴ Hence, the closing of the NMP domain is likely the rate-limiting step of the catalytic reaction. An early NMR study measured an opening rate for the combined NMP+LID domains of $286 \pm 85\text{ s}^{-1}$ ⁸⁴, and suggested that the rate-determining step is the enzyme's opening after the catalytic reaction.^{37,84} Possibly, this discrepancy arises from the simplistic two-state conformational transition model used in the NMR experiments for NMP+LID domain dynamics. A more complex model for AK domain dynamics has also been suggested by a more recent ^1H - ^{15}N HSQC NMR study, arguing that the chemical step is rate-determining.⁸⁵ However, also here, a two-state conformational transition model was used.

7) I think that a simplified figure that summarizes the finding of the allosteric effect can be added as an additional and final image linked to the discussion/conclusion section.

We are not sure what kind of figure the reviewer is thinking of. We added the allosteric events in Figure 3 and added a figure with the details of the AK active site in Figure 1.

Minor points:

1) Figure 1C would benefit from a schematic description of both Typ-a and Typ-b blockade. I believe only type-a is shown.

We added the reference to the Type-a and Type-b blockades.

2) In the third sentence of the introduction I think that “..with identical structures” should be replaced with “..with identical active-site structures” .

Done

3) In figure 1C the equilibrium between open and blocked nanopore can be quantified from the dwell times. Is this equilibrium far shifted towards the blocked state in the experiments? This can potentially be addressed from the equilibrium constant and the enzyme concentration used in experiments

In our experiment we use about 100 nM of AK and under these conditions a protein is most of the time inside a nanopore. We are not sure what equilibrium the reviewer is referring to. The dwell times indicate the time the protein is lodged inside the nanopore (typically a few seconds) and the time it takes for a second enzyme to enter once the first enzyme leaves the nanopore. These dwell times are enzyme specific as they depend on the electrostatic charge distribution, the shape and the size of the enzyme.

4) AMP has previously been proposed to inhibit AK by binding to the ATP site. In the current manuscript AMP inhibition is explained with an allosteric model. A discussion and comparison of both these models would be interesting.

The endosteric communication between the ligand in the active site might also explain the inhibitory effect observed at high concentrations of AMP. We observed M2 events in the presence of 1 mM AMP (Figure S6), suggesting that high AMP concentrations can trigger the closing of the LID domain, thus inhibiting the reaction. Possibly, LID closure is triggered when more than one AMP molecule binds to both the active sites.

We have added the connection between the allosteric effect and inhibition in the main text:

The ligand induced domain transitions of AK followed a well-defined hierarchy. In the absence of ATP, AMP was not observed to induce the closing of the NMP domain. By contrast, ATP induces the closing of the LID domain, and the subsequent arrival of AMP induces the closing of the NMP domain. These observations can be explained by the existence of an endosteric effect where the initial binding of AMP inhibits the binding of ATP by forming a less active conformation, as previously suggested,⁴⁹ or the ATP-induced closing of the LID domain induces the formation a higher-affinity binding site for AMP. In the M2-ternary-AMP:ATP:AK complex, the LID domain is more tightly closed (Figure 2I, right) than in the M2-binary-ATP:AK complex (LID opening of $118.9 \pm 36.3 \text{ s}^{-1}$ and $693.4 \pm 29.6 \text{ s}^{-1}$, respectively, Figure 2I, left). Since under these conditions AMP does not close the LID domain, the increased LID closing is the result of a second endosteric rearrangement.

5) The biological relevance of the findings are described a bit hand-waiving as a means to prevent water mediated hydrolysis of ATP and ADP. Are there other potential fitness benefits? Can the cellular concentrations of AMP and ATP be a clue?

In the conclusion we have proposed additional benefits:

In AK, both the evolution of allostery to control the binding sequence of ligands and the existence of conformers with varying AMP/ADP/ATP affinities, might enable fine-tuning of the enzyme's activity in response to changing cellular conditions or metabolic demands.

Reviewer #2 (Remarks to the Author):

The manuscript describes a study using a ClyA nanopore to confine adenylate kinase (AK) and investigate ligand-induced conformational changes of AK, manifested as ionic current fluctuations. Various ligands, such as the inhibitor Ap5A, ATP, AMP, and ADP, were tested. The

addition of each ligand, except for AMP, triggered current signal changes which were attributed to the ATP lid closed and ATP lid closed + AMP lid closed states. The data convincingly show that ATP binding is enhanced in the presence of AMP, demonstrating that AMP binding allosterically affects ATP lid closure. The experiments were well-designed, and the data analysis was carefully executed. The study provides concrete evidence of the allosteric effect by which the two ligand bindings regulate the lid domain movements. Directional lid domain closures, i.e., ATP lid closure preceding AMP lid closure, are supported by the observation that AMP alone did not induce any noticeable current signals. The conclusions are largely supported by the experiments. However, the manuscript can be improved in clarity, accuracy, and readability by addressing the following issues listed below:

1. The Catalytic Kinetic Model: In the proposed model, the closing of the NMP domain is suggested to be the rate-limiting step of the catalytic cycle. Does this observation agree with previous studies? Please discuss how the present study supports or argues against previous findings.

We have added a discussion about the rate-limiting step of the catalytic reaction.

The ability of to measure both LID and NMP domain motions independently revealed a precise hierarchy of domain closure during AK catalysis, controlled by endosteric interactions. The initial binding of ATP or ADP induces the closing of the LID domain at a rate of 1000 s^{-1} , which matches the domain closing rate that was measured by NMR for the combined NMP+LID domain motions ($1374 \pm 110\text{ s}^{-1}$).⁸⁴ The LID domain may quickly reopen with a similar rate (1000 s^{-1} , Figure 3). However, if another adenosine phosphate subsequently binds, it induces an endosteric effect that retards the opening of the LID domain and allows the additional closing of the NMP domain (Figure 2J). The closing of the NMP domain has the slowest observed rate (200 s^{-1}), which matches the catalytic turnover rate of the enzyme ($263 \pm 30\text{ s}^{-1}$).⁸⁴ Hence, the closing of the NMP domain is likely the rate-limiting step of the catalytic reaction. An early NMR study measured an opening rate for the combined NMP+LID domains of $286 \pm 85\text{ s}^{-1}$ ⁸⁴, and suggested that the rate-determining step is the enzyme's opening after the catalytic reaction.^{37,84} Possibly, this discrepancy arises from the simplistic two-state conformational transition model used in the NMR experiments for NMP+LID domain dynamics. A more complex model for AK domain dynamics has also been suggested by a more recent ¹H-¹⁵N HSQC NMR study, arguing that the chemical step is rate-determining.⁸⁵ However, also here, a two-state conformational transition model was used.

Notably, recent experimental and simulations findings^{51,58,85-88} have reported LID domain motions on the order of microseconds for both the apo- and liganded enzyme (a rate of $22,000\text{ s}^{-1}$,⁵¹). These large-scale LID-domain conformational dynamics, which might be required to position the substrates for their optimal orientation during catalysis,⁵¹ were not observed in our nanopore experiments. In this work, we used a higher KCl concentration (400 mM) to achieve a high signal-to-noise ratio. This could influence substrate binding and/or protein dynamics, since Cl⁻ ions compete with phosphates of the substrates in the binding pocket. Although a more physiological ionic concentration could also be used (Table S4), molecular modeling showed that under the experimental conditions (400 mM KCl and -80 mV) the concentration of Cl⁻ inside the nanopore is about 100 mM⁸⁹, that is similar to the concentration of Cl⁻ in vivo. Alternatively, nanopore confinement might have an influence on protein dynamics, but our previous work showed that this confinement has a minimal effect on the opening-closing rates for a range of proteins.^{90,91} In any case, the environment of the nanopore might be a better proxy for the crowded environment inside a cell compared to the highly diluted solution sampled by other single-molecule techniques. Instead, it is more likely that the high-frequency domain fluctuations are too fast for the sampling rates used in our experiments.

Although Mg^{2+} is required for the closing of the NMP domain, the network of endosteric interactions controlling the hierarchical closing of the domain is independent of Mg^{2+} . Structural data of the Ap5A closed conformation shows one Mg^{2+} ion binds to two phosphate groups of the ligand, while no direct interactions between Mg^{2+} and the residues in the LID or NMP domains are observed (Figure 1B). Domain closing is mediated by the binding of the catalytic residues R156, R123 (LID domain) and R36 (NMP domain)⁸⁶ to Ap5A (Figure 1B). It has been shown that Mg^{2+} positions the substrate optimally for catalysis and organizes the active site through indirect interactions via water molecules.^{29,35} This might directly affect the free energy barriers for the opening and closing of both the LID and NMP domains. In light of the data presented here, we argue that cooperative substrate and cofactor binding is likely to have evolved in AK for two reasons. Firstly, to facilitate efficient catalysis through proper substrate orientation. Secondly, to prevent unproductive enzyme closure in the absence of Mg^{2+} .⁸⁹

2. Figure 2c: The M1 state was interpreted as the AMP lid closed state. Please provide the reasoning for this interpretation.

M1 cannot be easily assigned as it is not directly correlated to the binding of any one ligand. Our reasoning is as follows (also added in the manuscript):

M1 cannot be easily assigned as it is not directly correlated to the binding of any one ligand. Likely, M1 reflects the closing of only one domain, the NMP domain, while the other domain, the LID domain, is still open. Interestingly, M1 to M2 blockades were often observed when both binding sites are occupied i.e.; when ATP and AMP or Ap5A are sampled, indicating that when both active sites are occupied the two domains cooperatively open and close.

Furthermore, we also tried different assignments for the M1 state in the kinetic model in Figure 3, but the only model that fitted the data well was for the M1 representing the closing of the NMP domain.

3. Figure 2E and 2G: In Figure 2E, the M2 signal is assigned to the ATP loaded + lid closed state, while in Figure 2G, M2 is shown as ATP loaded + AMP loaded + lid closed. This is confusing. The M2 state in Figure 2E should also be described as ATP loaded + AMP loaded + lid closed for consistency.

The reviewer is correct. The M2 signal corresponds to the ATP loaded + lid closed state AND to the ATP loaded + AMP loaded + lid closed state. However, under these conditions, the latter is more represented. We have added AMP to the figure.

The writing of the manuscript should be improved. It contains many words and phrases that are too vague to understand. Examples are listed below:

5. In the abstract: It states, "Here, we investigate the catalytic cycle of adenylate kinase (AK), an enzyme that catalyzes the interconversion of the various adenosine phosphates (ATP, ADP, and AMP)." This statement is incorrect. The manuscript does not investigate the catalytic cycle, which would focus on the transfer of the gamma-phosphate from ATP to AMP.

We agree with the reviewer that we do not measure AK catalysis directly, but we sample the conformational changes of the enzyme during catalysis. We have therefore changed into: "Here, we use single-molecule nanopore analysis to investigate the catalytic conformational changes of adenylate kinase (AK), an enzyme that catalyzes the interconversion of the various adenosine phosphates (ATP, ADP, and AMP)."

6. In the abstract: The authors state, "...that allosteric interactions enable converting ligands and cofactor binding energies into directional conformational changes of the two catalytic domains of AK." In biochemistry, allosteric interactions usually refer to interactions between the multi-subunits of oligomeric proteins. In AK, the binding of two ligands at different sites allosterically affects the lid and NMP's closure. The ligand interaction with AK at its active sites should not be called allosteric interactions..

This is an important point that the reviewer also brings in point 8. We agree with the reviewer that the transition is not as described in the textbook as allosteric transitions. We discuss this point jointly with the questions raised in point 8.

7. The first paragraph of the introduction: The content is irrelevant to the present study. This paragraph describes transition state theory and its application and limitations in explaining the enzyme's mechanism. However, the experiments in the manuscript do not study any aspect of the transition state of AK.

The reason for the initial paragraph is that transition state theory is used to explain the mechanism of enzymatic reactions. We show that enzymes provide a more complex framework than simply stabilizing the transition state of the reaction. Nonetheless, in light of this and reviewer 3 comments, we have extended the introduction with more mechanistic details about the reaction.

Introduction

Enzymes evolved to stabilize the transition state of a reaction. However, enzymes designed on the bases of transition-state stabilization have shown to capture only a fraction of the catalytic efficiency of natural enzymes.^{1,2} Additionally, transition state stabilization cannot explain why enzymes with identical active site structures show different catalytic efficiency at the same temperature,³ or why modifications far from the active site that do not affect the fold of the enzyme can have a large influence on catalysis.⁴⁻⁸ At the same time, it is now accepted that proteins are intrinsically 'dynamic'. However, the role of dynamics in enzyme catalysis remains a topic of heated discussion.⁹ It is possible, therefore, that enzyme structures and conformational dynamics evolved to have a more complex role than simply stabilize the transition state of a reaction.

In this work, we use single-molecule nanopore spectrometry¹⁰⁻¹⁹ to monitor the enzyme adenylate kinase (AK). Compared to other techniques such as single-molecule FRET, nanopore spectrometry allows label-free sampling of the entire enzyme's dynamics during multiple turnovers of individual enzymes and for minutes with microsecond resolution. AK catalyzes the reversible conversion of ATP and AMP to two molecules of ADP²⁰⁻²³, which is vital to maintain cellular energy homeostasis²³⁻²⁶. The enzyme consists of a rigid core domain that holds the active site, the ATP-binding LID domain, and the AMP-binding NMP-domain (Figure 1A)^{27,28}. The LID and NMP domains undergo major conformation changes mainly induced by the binding of ATP/ADP/AMP. R123, R156 in the LID domain and R36 in the NMP domain are key residues for ligand binding in the closed configuration (Figure 1B). Any mutation of these residues leads to substantially reduced enzyme activity.²⁹⁻³⁶ A Mg²⁺ cofactor ion is required for catalysis³⁷ and influence domain motion.³⁸ The role of magnesium, however, remains enigmatic because despite coordinating the two nucleotides enabling phosphate transfer,³⁹ it does not influence protein ligand affinity⁴⁰ nor interacts directly with either the NMP or LID domains in the closed state (Figure 1B).³⁵

Many studies have attempted to elucidate the role of domain motions in AK, and how dynamics and conformational changes are associated with molecular recognition, catalysis and allostery⁴¹⁻⁵⁸. It has been described that the binding of ATP/ADP/AMP induces the closing of the

domains. However, the nature of the motions of the two domains remains controversial, with some studies proposing that the LID domain closes first,^{52,54,55,58-70} others that the NMP domain closes first^{60,67,68,70,71} and others that both domain close simultaneously^{50,56,72}.

Our results showed that the LID and NMP domain close in a precise sequence, which allows regulating the enzyme's affinity and binding hierarchy for ATP, ADP, and AMP. A detailed kinetic analysis and modeling revealed a sophisticated mechanism in which the enzymatic function is regulated by multiple allosteric interactions that modulate the entire collection of enzyme dynamics.

8. On page 10: The authors wrote, "These observations suggest the existence of an endosteric effect whereas: the initial binding of AMP inhibits the binding of ATP, or the initial binding of AMP forms a less active conformation, as previously suggested, or the ATP-induced closing of the LID domain induces the formation of a higher-affinity binding site for AMP." How is the endosteric effect defined? What is the difference between this "endosteric" effect and the common biochemistry concept of "cooperativity"? Based on the author's description of the endosteric effect as AMP binding inhibiting ATP binding, it is just a classic negative cooperativity behavior. It seems that the authors invented a new term to replace a common term in the textbook, which decreases the readability of the manuscript.

This is an important point, and we agree we should not add a new terminology if not necessary. In biochemistry, allosteric regulation involves the binding of an effector molecule at a site other than the protein's active site. This binding induces conformational changes in the protein, affecting its activity.

Cooperativity refers to the behavior of enzymes or receptors with multiple binding sites. E.g. from the Encyclopedia Britannica: **Cooperativity**, in enzymology, a phenomenon in which the shape of one subunit of an enzyme consisting of several subunits is altered by the substrate (the substance upon which an enzyme acts to form a product) or some other molecule so as to change the shape of a neighbouring subunit. The result is that the binding of a second substrate molecule to the second subunit of the enzyme differs in strength or velocity from that of the first, the third from the second, and so on.

A famous example is the cooperative binding of O₂ to hemoglobin, with hemoglobine having 4 distinct binding sites for oxygen. The only example of a monomeric enzyme we found is Hexokinase IV, where the binding of glucose appears to be bimodal. However, the molecular mechanism is not understood, and cooperativity might be misleading as only one ligand (glucose) molecule binds to the active site, and not two ligands (as for ADP in AK). Hence, it is not clear how glucose can induce cooperative binding of glucose, if only one glucose molecule can bind to the active site.

The reviewer is then correct to point out that cooperativity is closely related to what we observe here (and to allostery). We used the term endostery because we observe cooperative/allosteric behavior between two molecules binding to the same binding site. Hence, this is a molecular mechanism in which two substrates affects each other binding affinity (in the same binding site). We suspect this behavior is common in enzymes, but more work has to be done.

9. On page 16: The authors mention, "where the collective interplay of the enzyme's soft interactions controls..." What do "soft" interactions mean? The term "soft" is not defined or explained in the manuscript.

We explained the term 'soft':

...soft interactions (e.g., the collective ensemble of the enzymes' interactions)

Reviewer #3 (Remarks to the Author):

I am commenting on the manuscript from the perspective of an expert in the study of conformational transitions (including allosteric communication) in various proteins, but I am not familiar with the relevance of the protein studied here - adenylate kinase (AK). However, I have searched the literature, and from the publications I have found, I believe that the authors have indeed addressed an open and relevant question and gained new insights. In particular, by using single-molecule nanopore spectrometry, they were able to show the sequence of ligand binding events (ATP, ADP and AMP) and how some of these events are controlled by allostery.

The experiments seem to have been performed with care. However, some of the analysis lacks diligence. The determination of the hidden Markov model seems to be in order, however the outcome of it is not surprising.

The manuscript is mostly clearly written.

In principle, the presented data could become publishable in Nature Communications. However, not all of the conclusions drawn are supported by the data. A number of issues need to be addressed before I can recommend this manuscript for publication. I list them in the order in which they appear in the manuscript, some of them are minor, others are important:

1) The methods used, i.e., nanopore spectrometry and hidden Markov modeling should be mentioned in the Abstract.

We have added the terms in the abstract.

2) Fig. 1A needs to be shown larger. ATP, AMP and Mg²⁺ need to be shown in brighter colors so that readers who are not that familiar with the system obtain a better understanding of the molecular details right at the start of the manuscript.

We have enlarged the figure and added an enlargement of the active site with a depiction of the main residues.

3) Following on from my comment 2), I found parts of the introduction not very informative. The first paragraph of the Introduction is very general and does not really have anything to do with the problem under investigation, while almost no molecular details about AK are given. Along with a more detailed look at Fig. 1A, I would like to know more about the key amino acids for ATP/ADP/AMP and Mg²⁺ binding, as this is important for understanding the binding events and their sequence. I will come back to this comment below in comment 10.

We have changed the introduction paragraph and adapted Fig. 1. We have added more molecular details to describe the interaction between the different domains and ATP/ADP/AMP

Introduction

Enzymes evolved to stabilize the transition state of a reaction. However, enzymes designed on the bases of transition-state stabilization have shown to capture only a fraction of the catalytic efficiency of natural enzymes.^{1,2} Additionally, transition state stabilization cannot explain why enzymes with identical active site structures show different catalytic efficiency at the same temperature,³ or why modifications far from the active site that do not affect the fold of the enzyme can have a large influence on catalysis.⁴⁻⁸ At the same time, it is now accepted that proteins are intrinsically 'dynamic'. However, the role of dynamics in enzyme catalysis remains a topic of heated discussion.⁹ It is possible, therefore, that enzyme structures and conformational dynamics evolved to have a more complex role than simply stabilize the transition state of a reaction.

In this work, we use single-molecule nanopore spectrometry¹⁰⁻¹⁹ to monitor the enzyme adenylate kinase (AK). Compared to other techniques such as single-molecule FRET, nanopore spectrometry allows label-free sampling of the entire enzyme's dynamics during multiple turnovers of individual enzymes and for minutes with microsecond resolution. AK catalyzes the reversible conversion of ATP and AMP to two molecules of ADP²⁰⁻²³, which is vital to maintain cellular energy homeostasis²³⁻²⁶. The enzyme consists of a rigid core domain that holds the active site, the ATP-binding LID domain, and the AMP-binding NMP-domain (Figure 1A)^{27,28}. The LID and NMP domains undergo major conformation changes mainly induced by the binding of ATP/ADP/AMP. R123, R156 in the LID domain and R36 in the NMP domain are key residues for ligand binding in the closed configuration (Figure 1B). Any mutation of these residues leads to substantially reduced enzyme activity.²⁹⁻³⁶ A Mg²⁺ cofactor ion is required for catalysis³⁷ and influence domain motion.³⁸ The role of magnesium, however, remains enigmatic because despite coordinating the two nucleotides enabling phosphate transfer,³⁹ it does not influence protein ligand affinity⁴⁰ nor interacts directly with either the NMP or LID domains in the closed state (Figure 1B).³⁵

Many studies have attempted to elucidate the role of domain motions in AK, and how dynamics and conformational changes are associated with molecular recognition, catalysis and allostery⁴¹⁻⁵⁸. It has been described that the binding of ATP/ADP/AMP induces the closing of the domains. However, the nature of the motions of the two domains remains controversial, with some studies proposing that the LID domain closes first,^{52,54,55,58-70} others that the NMP domain closes first^{60,67,68,70,71} and others that both domain close simultaneously^{50,56,72}. Our results showed that the LID and NMP domain close in a precise sequence, which allows regulating the enzyme's affinity and binding hierarchy for ATP, ADP, and AMP. A detailed kinetic analysis and modeling revealed a sophisticated mechanism in which the enzymatic function is regulated by multiple allosteric interactions that modulate the entire collection of enzyme dynamics.

4) On p. 4, the authors mention Type-a, Type-b and Type-c orientations of how AK enters the nanopore. They say that they focus on Type-a blockades. On p. 13, they talk about type II blockades. Are these then the Type-b blockades mentioned earlier? For these, they give a probability of 12+/-6%, but the probabilities of all other blockades are not given. And do the 12% mentioned on p. 13 correspond to the 10% mentioned on p. 17? This has to be reported with greater care and all details given. And what is with the Type-c / type III blockades?

We are sorry for the misunderstanding. Type a,b,c blockades are not type II blockades. Type II blockades are a subset of type-a blockades.

5) p. 7, 1st sentence: I guess that "that AMP increased the closed probability in the presence of AMP" should read "that AMP increased the closed probability in the presence of ATP". Please check.

We thank the reviewer for noticing this mistake.

6) p. 7: "When AMP was sampled in the presence of ATP" sounds strange. I think that "added" instead of "sampled" would be a better word here. Further below on that page, a similar such occurrence appeared, where "present" would be a better description of the situation.

We have changed the text according to the reviewer's suggestion.

7) Based on Figs. S9 and S10 the authors concluded that AMP did not induce the closing of the NMP domain. However, when I look at Fig. S9 I count 15 M1 events within 14 ms (and even 4

M2 events) and in Fig. S10 (no Mg²⁺) there are 7 M1 and 4 M2 events within 5 ms, whereas in Fig. 2A there are only 2 M1 events in > 600 ms. To me this seems that the addition of AMP greatly increased the closing of the NMP domain (and sometimes even causes the LID domain to close). Please explain why you interpret Figs. S9 and S10 quite differently than I do and conclude that M1 cannot exist when only AMP is added.

The reviewer is correct. Some events are observed. Initially, we thought these events might be due to ATP/ADP contaminants in AMP. However, it has been shown that AMP can induce the closing of AK domain(s). We think, therefore, that the most likely interpretation is that occasionally the binding of one or possibly two AMP molecules to AK induces the closing of the LID domain.

8) p. 10, 1st paragraph: Is the colon before “whereas” put on purpose there? And are the half-sentences after the colon thought as statements or possibilities? For the statement “the initial binding of AMP inhibits the binding of ATP” I could not find any data in the manuscript or SI. Please add! Without clear demonstration of this, the conclusions drawn may partly crumble.

We have corrected the mistake.

The AMP inhibitory effect on the binding of ATP has been described by a FRET study. We have simplified the text to make that connection more obvious.

The ligand induced domain transitions of AK followed a well-defined hierarchy. In the absence of ATP, AMP was not observed to induce the closing of the NMP domain. By contrast, ATP induces the closing of the LID domain, and the subsequent arrival of AMP induces the closing of the NMP domain. These observations can be explained by the existence of an endosteric effect whereas the initial binding of AMP inhibits the binding of ATP by forming a less active conformation, as previously suggested,⁴³ or the ATP-induced closing of the LID domain induces the formation a higher-affinity binding site for AMP. In the M2-ternary-AMP:ATP:AK complex, the LID domain is more tightly closed (Figure 2I, right) than in the M2-binary-ATP:AK complex ($118.9 \pm 36.3 \text{ s}^{-1}$ and $693.4 \pm 29.6 \text{ s}^{-1}$, respectively, Figure 2I, left). Since under these conditions AMP does not close the LID domain, the increased affinity of ATP for AK is the result of a second endosteric rearrangement.

9) p. 10 / Fig. 3: The reader has to guess what the star(s) at states Mx mean. My interpretation is that these are the hidden states where ATP/AMP has bound but the conformational transitions have not occurred yet. However, when looking at Fig. 3, this does not apply to M2* and M3** as these are closed states which can be detected by nanopore spectrometry. And why have states M2 and M3 vanished altogether in Fig. 3? What are the numbers at the arrows: times, rates, concentrations? Units would be great and a better notation - it took me a while to understand what the “E” meant.

The star indicate the binding of one (*) or two (**) ligands to AK. We have now specified it in the main text

A side comment is in place: The authors assume that AxP binding precedes the conformational changes; yet this is not shown here and might not be the case; these could also be concerted AxP binding events!

10) The Conclusions would greatly benefit from adding a discussion / interpretation of the findings on a more detailed structural level (amino-acid level) to rationalize them (see also my comment 3).

We have added the discussion as suggested by the reviewer.

11) p. 27: Please explained the meaning of "y" and what "pdf" stands for.

"y" is a vector that contains the time series of current level measurements from a particular experiment. The measurement vectors from different experiments (y_1, y_2, \dots) were used together to fit the hidden Markov model. The descriptions of the individual experiments can be found in the subsection "Data selection and preparation" of Materials and Methods. "Pdf" is an abbreviation of the probability density function. We made these terms clearer in the text.

Reviewer #3 (Remarks on code availability):

I reviewed the theory section accompanying the code. The theory seems to be okay, but not all symbols were explained, like the meaning of "y" or what "pdf" stands for.

The code produced the expected outcome; any other predicted Markov model would have raised my concerns.